# Gut insulin action protects from hepatocarcinogenesis in diabetic mice comorbid with nonalcoholic steatohepatitis

Kotaro Soeda [1,2], Takayoshi Sasako [1,2], Kenichiro Enooku[3], Naoto Kubota[2], Naoki Kobayashi[1], Yoshiko Matsumoto Ikushima[1], Motoharu Awazawa[1], Ryotaro Bouchi[4], Gotaro Toda[1,2], Tomoharu Yamada[3], Takuma Nakatsuka [3], Ryosuke Tateishi [3], Miwako Kakiuchi[5], Shogo Yamamoto [5], Kenji Tatsuno [5], Koji Atarashi [6,7], Wataru Suda[7], Kenya Honda [6,7], Hiroyuki Aburatani [5], Toshimasa Yamauchi [2], Mitsuhiro Fujishiro [3], Tetsuo Noda[8], Kazuhiko Koike[3], Takashi Kadowaki[2,9,10] & Kohjiro Ueki [1,11] ✉

Diabetes is known to increase the risk of nonalcoholic steatohepatitis (NASH) and hepatocellular carcinoma (HCC). Here we treat male STAM (STelic Animal Model) mice, which develop diabetes, NASH and HCC associated with dysbiosis upon low-dose streptozotocin and high-fat diet (HFD), with insulin or phlorizin. Although both treatments ameliorate hyperglycemia and NASH, insulin treatment alone lead to suppression of HCC accompanied by improvement of dysbiosis and restoration of antimicrobial peptide production. There are some similarities in changes of microflora from insulin-treated patients comorbid with diabetes and NASH. Insulin treatment, however, fails to suppress HCC in the male STAM mice lacking insulin receptor specifically in intestinal epithelial cells (ieIRKO), which show dysbiosis and impaired gut barrier function. Furthermore, male ieIRKO mice are prone to develop HCC merely on HFD. These data suggest that impaired gut insulin signaling increases the risk of HCC, which can be countered by restoration of insulin action in diabetes.

Epidemiologically, diabetes is reported to be an independent risk factor for hepatocellular carcinoma (HCC)[1]. Diabetes is also one of the major risk factors for non-alcoholic steatohepatitis (NASH)[2] leading to the development of hepatocellular carcinoma (HCC). Although some antidiabetic drugs have been suggested to inhibit NASH in patients with diabetes or prediabetes[3], it remains unclear whether any specific strategy in diabetes treatment may lead to the inhibition of HCC, mainly because the pathophysiological mechanism of NASH and HCC associated with diabetes is not fully understood.

[1]Department of Molecular Diabetic Medicine, Diabetes Research Center, Research Institute, National Center for Global Health and Medicine, Tokyo, Japan. [2]Department of Diabetes and Metabolic Diseases, Graduate School of Medicine, The University of Tokyo, Tokyo, Japan. [3]Department of Gastroenterology, The University of Tokyo, Tokyo, Japan. [4]Diabetes and Metabolism Information Center, Diabetes Research Center, Research Institute, National Center for Global Health and Medicine, Tokyo, Japan. [5]Genome Science Division, The University of Tokyo, Tokyo, Japan. [6]Department of Microbiology and Immunology, Keio University School of Medicine, Tokyo, Japan. [7]RIKEN Center for Integrative Medical Sciences, Yokohama City, Kanagawa, Japan. [8]Department of Cell Biology, Cancer Institute, Japanese Foundation of Cancer Research, Tokyo, Japan. [9]Department of Prevention of Diabetes and Lifestyle-Related Diseases, Graduate School of Medicine, The University of Tokyo, Tokyo, Japan. [10]Toranomon Hospital, Tokyo, Japan. [11]Department of Molecular Diabetology, Graduate School of Medicine, the University of Tokyo, Tokyo, Japan. ✉e-mail: uekik@ri.ncgm.go.jp

Type 2 diabetes is a syndrome characterized by chronic hyperglycemia due to decreased insulin action in target organs as a consequence of insulin resistance and insufficient compensation of insulin secretion. Insulin-secretory capacity against insulin resistance differs among ethnic groups[4, 5]. Indeed, Asian subjects have been shown to have a much lower insulin-secretory capacity and develop diabetes at a significantly lower body mass index (BMI) and through much less severe insulin resistance than Caucasian subjects, with the BMI of Asian subjects comorbid with diabetes and NASH/HCC also shown to be lower than that in their Caucasian counterparts. Therefore, insulin treatment is often required for patients with diabetes and liver damage, given that many oral antidiabetic medications are not readily available for use in these patients due to associated increases in the risk of adverse reactions. On the other hand, there is a controversy over the use of insulin for patients with diabetes at risk of cancer, as it mediates not only the glucose-lowering but the cell proliferative effect of insulin receptor signaling, thus possibly leading to the outgrowth of latent tumor buds. In addition, previous studies have demonstrated that hepatocyte-specific disruption of insulin receptor substrates in mice suppresses hepatocarcinogenesis induced by diethylnitrosoamin (DEN) on high-fat diet feeding[6], suggesting that insulin action in the liver may be required for the development of chemically-induced HCC. Moreover, the Warburg effect, which is often induced in carcinoma cells, might be promoted by insulin signaling via activation of hypoxia-inducible factor-1 (HIF-1)[7], leading to a tumor-favorable microenvironment. However, how the systemic effect of insulin as it is mediated by multiple organs may have a role in this pathophysiology remains poorly understood. Therefore, it is necessary from a systemic pathophysiological viewpoint to determine what kind of diabetes treatment is suitable for patients with diabetes and NASH to prevent the progression of NASH and the development of HCC.

Recently, new light has been shed on the gut microbiome as a factor influencing systemic metabolism and various diseases, including diabetes, obesity, NASH, and HCC[8]. The composition of gut microbiota is influenced by the barrier function of the host, which plays a role in preventing the invasion of chemical poisons, viruses, and microorganisms. The mutual interaction between host and microbes are closely related to host's internal condition, such as nutrient intake[9], obesity[10], intestinal inflammation[11] and oral medications[12]. For instance, previous studies suggested that intestinal epithelial insulin receptor signaling might be supportive for intestinal Paneth cells, which secrete antimicrobial peptides (AMPs) to maintain chemical barrier function against invasive microorganisms[13].

Here, we demonstrate the protective role of insulin signaling in the gut against hepatocarcinogenesis associated with maintaining intestinal barrier function and suppressing dysbiosis using diabetic NASH-HCC model mice and samples from patients with diabetes and NASH.

## Results

### STAM mouse is a model of NASH-HCC associated with diabetes

Recently, a growing number of NASH patients develop HCC[14], and diabetes is shown to be a common risk factor for these diseases[1]. It is also known that Asian NASH patients show relatively modest obesity[15], unlike Caucasian patients, presumably due to the fact that Asian subjects exhibit a lower compensational insulin-secretory capacity against insulin resistance associated with steatohepatitis than Caucasian subjects exhibiting marked obesity and hyperinsulinemia. Of the various mouse models available for human NASH and HCC research, STAM mice (STelic Animal Model mice) represent a non-obese diabetic NASH-HCC model, which develop hepatic fibrosis and HCC at 9 weeks and 20 weeks of age, respectively[16] (Fig. 1a). Following the protocol, STAM mice were generated by given mice a small-dose (200 μg/head) subcutaneous streptozotocin (STZ) injection on birth and high-fat diet (HFD) feeding from their weaning[16] (Fig. 1a).

STAM mice showed hyperglycemia (Fig. 1b), dyslipidemia, and hepatic steatosis, although they were associated with less hyperinsulinemia to the level of normal mice without HFD feeding (Fig. 1c), obesity (Supplementary Figure 1a, Supplementary Fig. 1b) and hepatic steatosis than high-fat-diet-fed obese mice without STZ injection (diet-induced obese [DIO] mice).

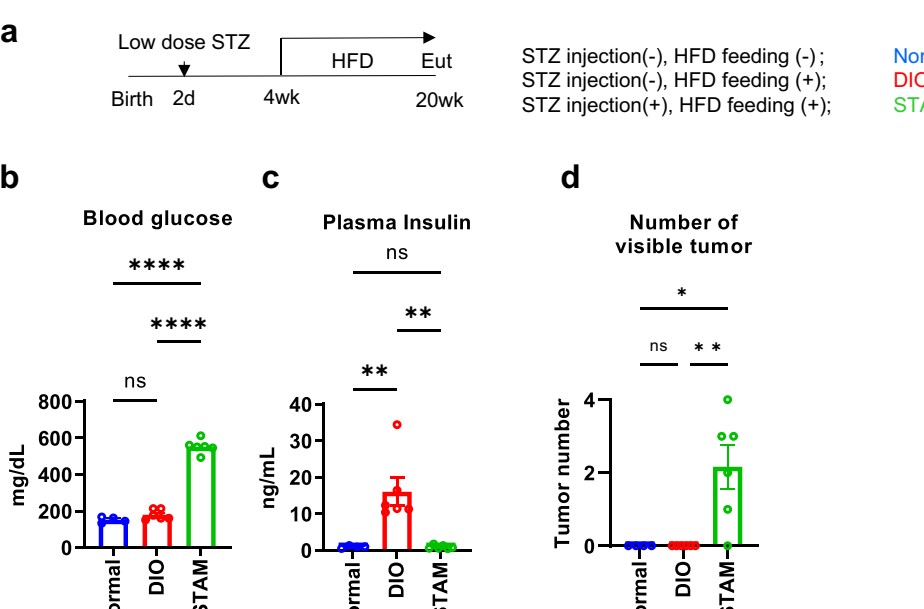

**Fig. 1 | Generation and characteristics of STAM mice as diabetic NASH-HCC model.** The experiment protocol (**a**). The blood glucose (**b**) and plasma insulin (**c**) of normal lean mice (Normal, *n* = 4), diet induced obese mice (DIO, *n* = 6), and STAM mice (*n* = 6) (**b, c**). Values of the data are expressed as mean ± SEM. **P < 0.01, ****P < 0.0001. One-way ANOVA, with Tukey's multiple comparisons test. The number of visible tumors (**d**) of normal lean mice (Normal, *n* = 4), diet induced obese mice (DIO, *n* = 6), and STAM mice (*n* = 6) (**d**). Values of the data are expressed as mean ± SEM (**b, c, d**). *P < 0.05, **P < 0.01. One-way ANOVA, with Dunn's multiple comparisons test (**b, c, d**). Source data are provided as a Source Data file. The exact P values are provided in Supplementary Data 3.

Consistently with previous studies, fibrosis was evident in STAM mice compared to that in DIO mice, and the hepatic mRNA expression of *Collagen 1* was elevated in STAM mice (Supplementary Fig. 1c) with hepatic fibrosis also confirmed by Sirius red staining (Supplementary Fig. 1d). One of the most important characteristics of STAM mice was that they were associated with a high rate of hepatocarcinogenesis (Fig. 1d, Supplementary Fig. 1d). In relation to hepatic fibrosis and hepatocarcinogenesis, the expression of hepatic inflammatory cytokines (i.e., *Tnf-α*, *Ccl2*, and *Il-6*) was elevated in STAM mice (Supplementary Fig. 1c), compared with DIO mice. Multiple somatic mutations induced by STZ were found not only in hepatic tumor regions (Supplementary Fig. 1e, f) but also in non-tumor areas of the STAM mouse liver (Supplementary Fig. 1g). The signatures of the mutations were consistent with those induced by alkylating agents (Supplementary Fig. 1h) and were shown to be closest to those of human NASH/HCC among several animal models[17]. These data suggest that certain somatic mutations caused by STZ treatment on birth had been maintained by cell proliferation in STAM mice until HCC became evident.

Therefore, the characteristics of this animal model may prove helpful in investigating the pathogenesis of non-obese diabetes comorbid with NASH/HCC as well as in exploring treatment strategies suitable for patients with these diseases.

### Insulin treatment, but not phlorizin treatment, led to suppression of hepatocarcinogenesis in STAM mice

Insulin is widely used and sometimes the only option for patients with diabetes and liver dysfunction, while there is a concern that insulin might increase the risk of cancer through its mitogenic effects. On the other hand, recent studies have suggested that inhibition of sodium-glucose cotransporters (SGLTs), such as SGLT2 inhibitors, which lower blood glucose levels through the promotion of urinary glucose excretion without increasing insulin levels, may offer a beneficial effect on NASH through inhibition of hepatic lipid accumulation. To assess the effects of these two different glucose-lowering approaches on the development of diabetes-related NASH/HCC, STAM mice were treated with insulin analogue (insulin glargine) or phlorizin (PHZ), an SGLT1/SGLT2 inhibitor, and compared with non-treated control STAM mice (Fig. 2a, Supplementary Fig. 2a, b).

While insulin signaling is known to promote lipid synthesis in the liver, reductions were seen in this study in the pathological grade of NASH, hepatic steatosis (Fig. 2b, c, Supplementary Fig. 2c–e), fibrosis (Supplementary Fig. 2f), and hepatocarcinogenesis (Fig. 2d, e) in insulin-treated STAM mice when they were 20 weeks of age. On the other hand, NASH was significantly reduced in pathological grade in the PHZ-treated STAM mice compared to that in non-treated STAM mice (Fig. 2b, c, Supplementary Fig. 2c–f), as was observed in previous studies[18,19] in which SGLT2 inhibitors were used. However, the expression level of *Col1* was not significantly reduced, and the number and maximum diameter of tumors were not significantly improved (Fig. 2d–f). Compared with control STAM mice, a significantly higher level of plasma insulin was observed in insulin-treated STAM mice, while no significant induction was detected in PHZ-treated STAM mice, although both treatments ameliorated hyperglycemia significantly (Fig. 2g, h).

Given the results from insulin-treated and PHZ-treated mice, it was suggested that insulin signaling has certain protective effects against the development of HCC as well as hepatic steatosis and fibrosis, besides its glucose-lowering effects.

### Insulin treatment suppressed NASH in STAM mice at multiple steps

To explore the mechanism by which insulin suppressed the progression of NASH and HCC in STAM mice, gene expression analyses were performed for steatosis and fibrosis in these mice at 9 weeks of age,

following on from a previous study, which suggested that gene expression profiles in steatosis showed a wide divergence by the time HCC become evident[16]. At this point, body weight and liver weight were not significantly different in insulin- or PHZ-treated STAM mice from those in non-treated STAM mice (Supplementary Fig. 3a, b). Hepatic accumulation of triglyceride (Supplementary Fig. 3c) was suppressed by treatment with insulin or PHZ, while that of cholesterol was not suppressed significantly (Supplementary Fig. 3d).

The development of hepatic steatosis may likely involve the following three possible mechanisms: 1) increased lipid synthesis in the liver; 2) increased lipid transport to the liver; and 3) decreased lipid oxidation in the liver. The expression of fatty acid synthase (FAS) and acetyl-CoA carboxylase (ACC), rate limiting enzymes of de novo lipogenesis in the liver, is known to be regulated by insulin signaling[20], and indeed the expression of *Fas* was shown to be upregulated in insulin-treated STAM mice, but both *Fas* and *Acc* were not significantly different compared to that in high-fat-diet-fed mice without STZ injection (Supplementary Fig. 3e), suggesting that the regulation of de novo lipid synthesis in the liver may not play a major role in the development of hepatic steatosis in STAM mice or its amelioration by insulin treatment. Next, we investigated the factors which play roles in lipid influx from circulating blood to the liver. Insulin treatment significantly suppressed plasma NEFA concentration and loss of epididymal white adipose tissue (eWAT) mass (Supplementary Fig. 3f, g). In eWAT, phosphorylated hormone-sensitive lipase (HSL) was also upregulated in control STAM mice but was suppressed by insulin treatment (Supplementary Fig. 3h). These data suggest that insulin supplementation preserved the fat-storing function of WAT to suppress circulating fatty acids and promote their incorporation into the liver in STAM mice. Consistent with a previous study[21], the incorporation of plasma NEFA to the liver might be important for hepatic steatosis in this model mice as also shown in humans. In eWAT, *Ucp1* was also induced in STAM mice and reduced by insulin treatment (Supplementary Fig. 3i), suggesting that lipid combustion was also involved in the reduction of eWAT mass. The hepatic expression of fatty acid transporters (*Cd36* and *Fatp2*) and perilipins, a key regulator of lipid droplet formation, were upregulated in control STAM mice, and suppressed by insulin treatment (Supplementary Fig. 3j). Significant reductions were seen in *Cyp7b1* and *Cyp27a1* expression, while expression of *Cyp7a1* and *Cyp8b1* did not change significantly in the STAM model compared with DIO (Supplementary Fig. 3j).

On the other hand, PHZ treatment did not suppress the level of pHSL in WAT (Supplementary Fig. 3h), the plasma level of NEFA (Supplementary Fig. 3g), and the hepatic expression level of *Cidec*, *Plin2*, *Plin3*, or *Cd36* (Supplementary Fig. 3j). Another key transport molecule, *Fatp2*, was induced in the STAM model and suppressed by treatment of insulin or PHZ, suggesting its involvement in effects of both treatments.

These data suggest that insulin ameliorates hepatic steatosis by preserving fat storage in white adipocytes under diabetic conditions, consistent with previous clinical studies[22].

Real-time PCR analyses also provided evidence for the upregulation of expression of *Col1* in the liver of STAM mice, another important characteristic of NASH (Supplementary Fig. 1c). Insulin treatment led to amelioration of fibrosis with reduction of *Col1* (Fig. 2f). PHZ treatment also suppressed the Sirius red positive area in the liver of STAM mice (Supplementary Fig. 2f), but did not suppress the hepatic expression of *Col1* mRNA (Fig. 2f). These data suggest that insulin signaling promoted tight suppressive regulation of collagen production in STAM mice.

We also investigated the expression of inflammatory cytokines. *Ccl2*, *Tnfa*, and *Il6* were significantly suppressed by insulin treatment (Fig. 2i), unlike in PHZ-treated STAM mice, suggesting a protective role of insulin signaling against chronic hepatic inflammation.

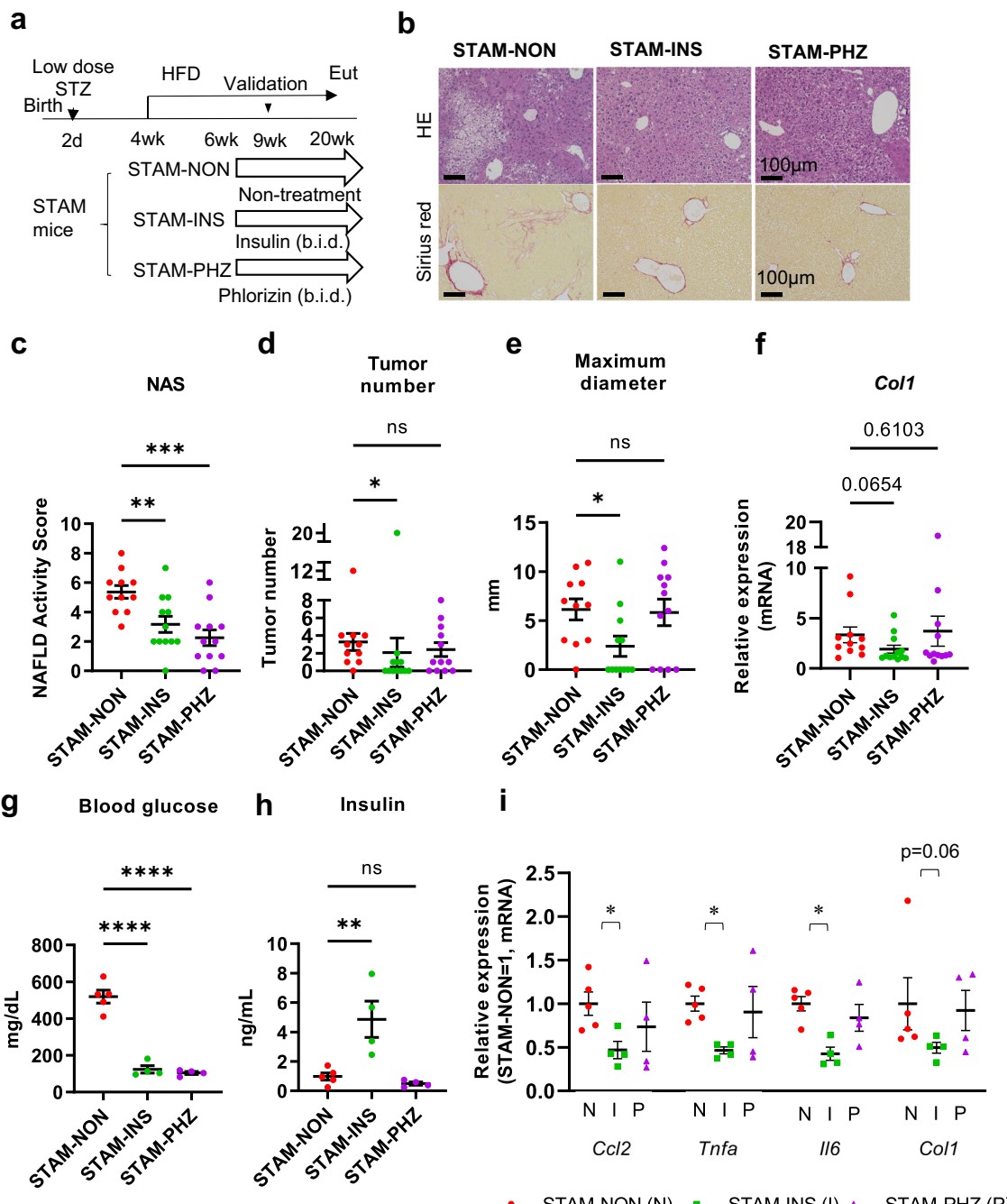

**Fig. 2 | Suppression of hepatocarcinogenesis by insulin treatment in STAM mice.** The protocol of the drug intervention experiment (**a**). Microscopic appearance of liver. HE staining and Sirius red staining (**b**). Scale bar, 100 μm. NAFLD activity score (NAS) assessed by HE staining (**c**). **$P < 0.01$, ***$P < 0.001$. One-way ANOVA, with Dunnett's multiple comparisons test. Tumor number (**d**) and maximum diameter (**e**) of mice at 20 weeks of age. *$P < 0.05$. One-way ANOVA, with Dunn's multiple comparisons test (**d**) and Dunnett's T3 multiple comparisons test (**e**) respectively. Relative mRNA expression level of *Col 1* in liver at 20 weeks of age (**f**). Non-treated STAM mice (STAM-NON) $n = 11$, insulin-treated STAM mice (STAM-INS) $n = 12$, phlorizin-treated STAM mice (STAM-PHZ) $n = 12$ (**c, d, e, f**). The expression level of *Col 1* is presented by ratio to normal lean mice ($n = 3$, **f**). The *P* values are provided by one-way ANOVA, with Dunn's multiple comparisons test (**f**). Blood glucose (**g**) and plasma insulin level (**h**) of mice at 9 weeks of age. **$P < 0.01$, ****$P < 0.0001$. One-way ANOVA, with Dunnett's multiple comparisons test. Relative expression level in liver at 9 weeks of age (**i**). STAM-NON $n = 5$, STAM-INS $n = 4$, STAM-PHZ $n = 4$ (**g, h, i**). The expression level is presented by ratio to STAM-NON mice (**i**). *$P < 0.05$. Unpaired 2-tailed *t*-test. Values of the data are expressed as mean ± SEM (**c, d, e, f, g, h, i**). Source data are provided as a Source Data file. The exact *P* values are provided in Supplementary Data 3 unless they are below 0.0001.

## Insulin treatment suppressed cell proliferation pathway in the liver of STAM mice

While insulin suppressed the progression of NASH in the liver of STAM mice to a similar extent with PHZ, through distinct mechanisms, insulin alone suppressed the development of HCC. We therefore tried to explore how insulin suppressed hepatocarcinogenesis in STAM mice.

Gene ontology analysis (Metascape, version 3.0[23]) (Supplementary Fig. 4a) suggested that cell division (GO 0051301; Log *P* value = −14) and regulation of mitotic cell cycle (GO 0007346; Log *P* value = −6.5) were upregulated in the liver of STAM mice, and GO TRRUST (version 2) analysis[24] (Fig. 3a) suggested that Warburg effect was enhanced in the liver of STAM mice and that Hif-1α represented a

**a**

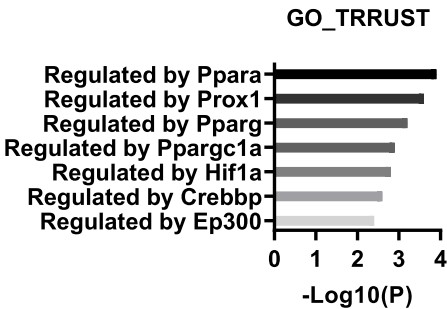

**b**

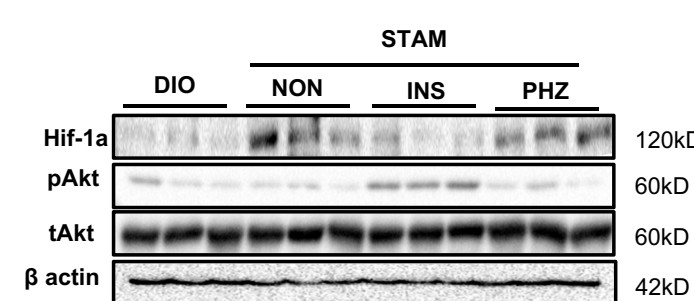

**c** **d** **e** **f**

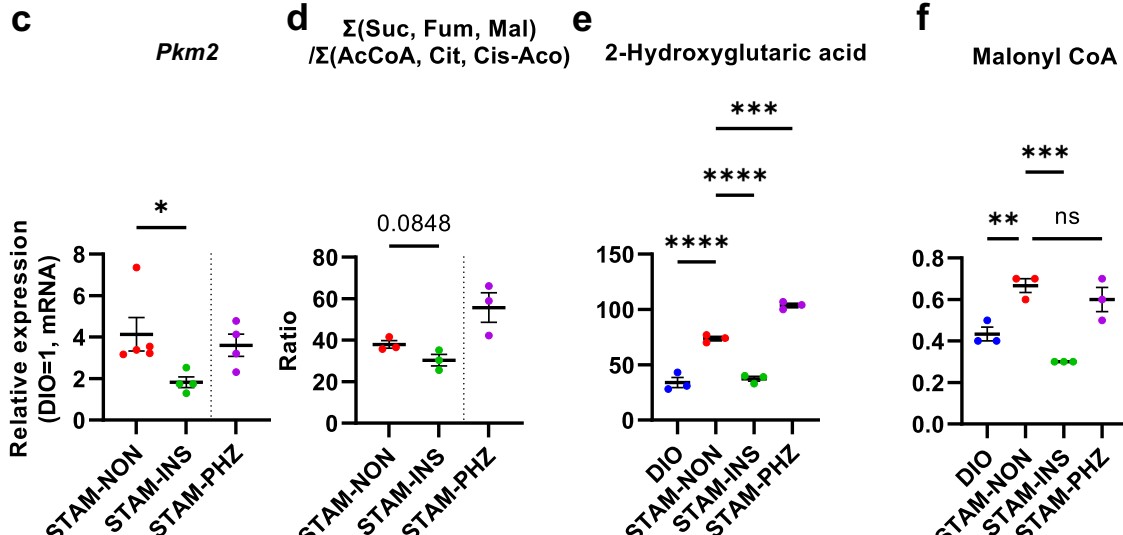

**Fig. 3 | Suppression of Warburg effect by insulin treatment in STAM mice.** GO TRRUST analysis performed through microarray analysis of 3 groups (**a**, $n = 3$ for each group). Western blotting analysis of liver tissue at 9 weeks of age (**b**). The experiments were repeated independently at least twice. Relative mRNA expression level of *Pkm2* in liver at 9 weeks of age (**c**). Non-treated STAM mice (STAM-NON) $n = 5$, insulin-treated STAM mice (STAM-INS) $n = 4$, phlorizin-treated STAM mice (STAM-PHZ) $n = 4$, $*P < 0.05$. 2-sided unpaired $t$ test (**c**). The index calculated by hepatic accumulation of succinate (Suc), fumarate (Fum), malate (Mal), acetyl CoA (AcCoA), citrate (Cit), cis-aconitic acid (Cis-Aco) (**d**). The values are presented by ratio. DIO $n = 3$, STAM-NON $n = 3$, STAM-INS $n = 3$, STAM-PHZ $n = 3$, assessed as representative samples by CE-TOF/MS, Hepatic accumulation of 2-Hydroxyglutaric acid (**e**). Hepatic accumulation of malonyl CoA (**f**). DIO $n = 3$, STAM-NON $n = 3$, STAM-INS $n = 3$, STAM-PHZ $n = 3$, assessed as representative samples by CE-TOF/MS, $**P < 0.01$, $***P < 0.001$, $****P < 0.0001$, one-way ANOVA, with Dunnett's multiple comparison test (**e**, **f**). Values of the data are expressed as mean ± SEM (**c**, **d**, **e**, **f**). Source data are provided as a Source Data file. The exact $P$ values are provided in Supplementary Data 3 unless they are below 0.0001.

candidate upstream regulator. Indeed, major downstream factors, *Pkm2* and *Gls1* of Hif-1α, were upregulated in STAM mice (Fig. 4b, Supplementary Fig. 4b, c). Metabolomics analysis (CE-TOF/MS) also revealed that the ratio of the sum of the latter half (succinate, fumarate, malate) of the TCA cycle to that of the former half (acetyl CoA, citrate, cis-aconitic acid) was significantly upregulated in the liver of STAM mice (Supplementary Fig. 4d, e), consistent with the existence of the Warburg effect.

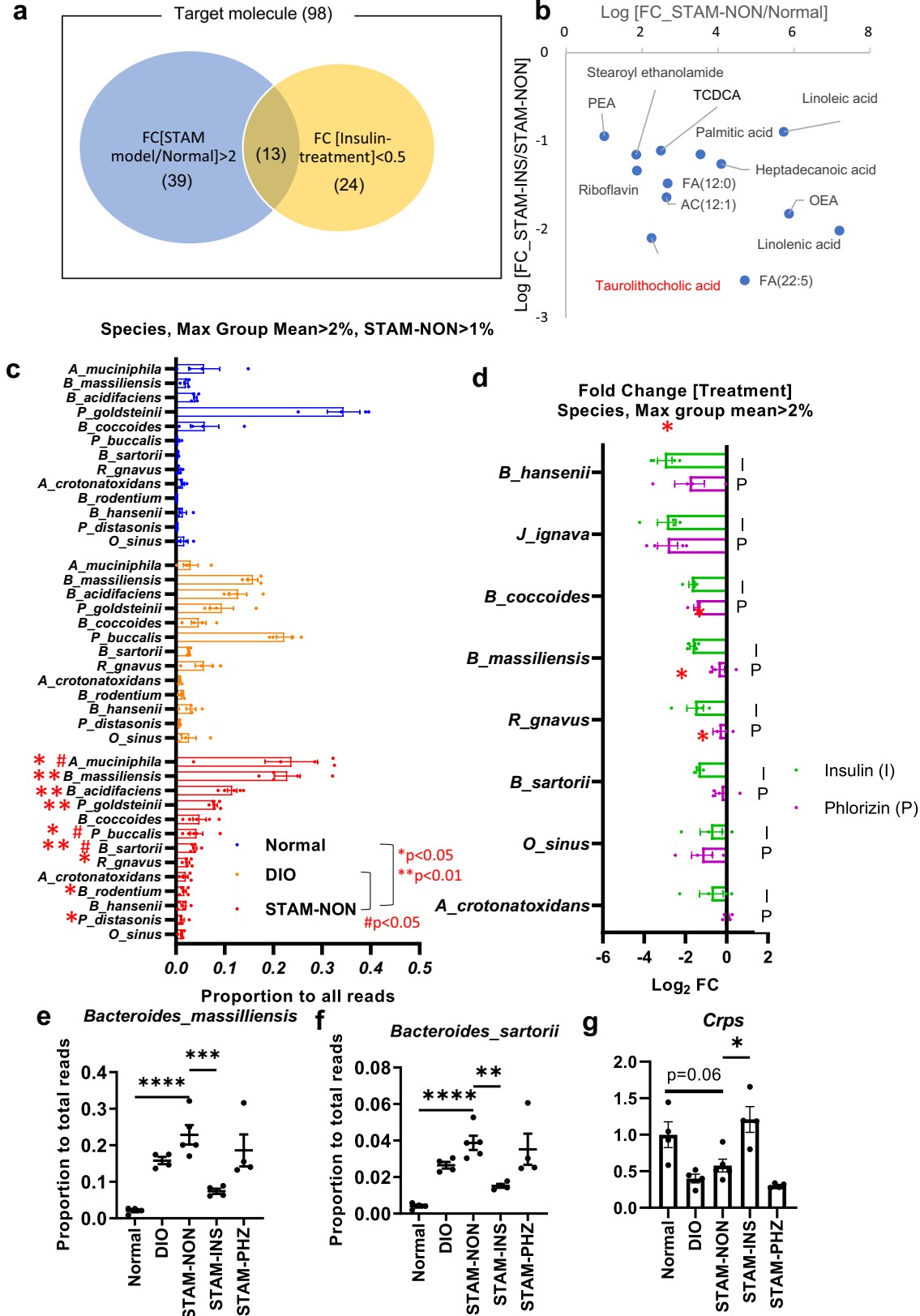

Along with these findings, increased protein expression of Hif-1α was already detected by Western blotting (Fig. 3b), and *pkm2* mRNA was also upregulated in non-treated STAM mice even at 9 weeks of age before HCC became evident (Fig. 3c). Metabolomic analysis showed that insulin treatment reduced the ratio of the metabolites [Σ(Suc, Fum, Mal)/Σ(AcCoA, Cit, Cis-Aco)] and glutamine (Fig. 3d,

Supplementary Fig. 4f), suggesting that the Warburg effect was present in STAM mice but suppressed by insulin treatment.

Primary component analysis revealed the comprehensive metabolic differences in CE-TOF/MS among treatment groups (Supplementary Fig. 5). Specifically, the metabolomic analysis revealed that 2-hydroxyglutaric acid (2-HG, Fig. 3e) and malonyl CoA (Fig. 3f) were

**Fig. 4 | Attenuation of impairment of intestinal barrier function and dysbiosis of STAM mice by insulin treatment.** A Venn diagram of metabolites in liver assessed by LC/MS analysis (**a**). The number of metabolites which were more abundant in non-treated STAM mice (STAM model, $n = 3$) than in normal mice (Normal, $n = 3$) (52 metabolites) and metabolites more abundant than in insulin-treated STAM mice ($n = 3$) (Insulin treatment; 37 metabolites) (threshold fold change, 2.0). 13 metabolites met both criteria (**a**). 13 metabolites condensed were plotted by each fold change. Secondary bile acids are presented by red letters (**b**). Proportion of bacteria to total reads in each sample in normal lean mice ($n = 4$), DIO mice ($n = 4$), STAM mice ($n = 5$) without treatment (**c**). The bacteria were selected by max group mean, which was more than 2% and the read proportion in non-treated STAM mice (STAM-NON) was more than 1%. *$P < 0.05$, **$P < 0.01$,

2-sided unpaired t test, normal vs STAM-NON, # $P < 0.05$, two-sided unpaired t test, DIO vs STAM-NON (**c**). Fold change by treatment ($n = 4$, insulin or phlorizin, **d**). *$P < 0.05$, 2-sided unpaired t test, STAM-NON vs STAM-INS, or STAM-NON vs STAM-PHZ (**d**). The bacteria suppressed by treatment were selected. The proportion to total reads of *Bacteroides massiliensis* (**e**), *Bacteroides sartorii* (**f**). qPCR analysis of *Crps* (**g**). Normal lean mice ($n = 4$), DIO mice ($n = 4$), STAM-NON ($n = 5$), insulin-treated STAM mice (STAM-INS, $n = 4$), phlorizin-treated STAM mice (STAM-PHZ, $n = 4$), *$P < 0.05$, **$P < 0.01$, ***$P < 0.001$, ****$P < 0.00001$, one-way ANOVA, with Dunnett's multiple comparisons test (**e, f, g**). Values of the data are expressed as mean ± SEM (**c, d, e, f, g**). Source data are provided as a Source Data file. The exact $P$ values are provided in Supplementary Data 3 unless they are below 0.0001.

accumulated in STAM mice and suppressed by insulin-treatment. 2-HG is known as an oncometabolite[25–27], and malonyl CoA could act as a substrate of lipogenesis, respectively.

### Insulin suppressed the accumulation of potently oncogenic bile acids associated with changes in the microbiome in STAM mice

In the liver of STAM mice, the Liquid Chromatography-Mass Spectrometry (LC/MS) analysis revealed that 13 metabolites were upregulated in STAM mice compared with those in normal and insulin-treated STAM mice (Fig. 4a, Supplementary Table 1, see **Methods** for details, Supplementary Fig. 6a for PCoA), which included taurolithocholic acid (secondary bile acid) (Fig. 4b). In the LC/MS analysis, the ratio of deoxycholic acid to cholic acid (DCA/CA) increased in STAM mice and was suppressed by insulin treatment (Supplementary Fig. 6b).

A previous report on obesity-related HCC revealed that secondary bile acids produced by dysbiotic flora promoted HCC initiation via cellular senescence[8]. Therefore, the 16 S metagenomic analyses were performed using fecal DNA samples of normal chow-fed mice, diet-induced obese mice (DIO mice) without STZ treatment, non-treated, insulin-treated and PHZ-treated STAM mice. The PCoA revealed differential components of gut microbiota between STAM mice and DIO mice (Supplementary Fig. 6c), and that the differences in gut flora between non-treated STAM mice and DIO mice were partially preserved by insulin treatment, but not by PHZ treatment (Supplementary Fig. 6c). Assessment of individual bacteria revealed that *Bacteroides massiliensis* was the second most abundant species in non-treated STAM mice (Fig. 4c, Supplementary Table 2), and the proportion of this bacterium was significantly suppressed by insulin treatment (Fig. 4d, e, Supplementary Table 3). The proportion of *Bacteroides sartorii*, the third most abundant species (Fig. 4c) and genetically close to *B. massiliensis*, exhibited a similar profile (Fig. 4d, f)[28]. On the other hand, the proportions of *B. massiliensis* and *B. sartorii* were not suppressed by PHZ treatment (Fig. 4d–f). The relevance of the data from PCoA and individual analyses suggest that these bacteria could be a good marker for dysbiosis in this model mice. The bacterium was reported to have bile salt hydrolase (BSH) responsible for the deconjugation of taurine- or glycin-conjugated primary bile acids, based on the database (PATRIC 3.6.12, Bacteroides massiliensis B84634 = Timone 84634 = DSM 17679 = JCM 13223), and thus could be involved in supplying a substrate for secondary bile acids production, although it has not been experimentally confirmed because of the difficulty in inoculation of the bacteria.

Of the factors responsible for intestinal chemical barrier function which could influence gut flora, *Crps* (Cryptdins, α-defensins) was upregulated by insulin treatment in the ileum, which was not the case with PHZ treatment (Fig. 4g). Furthermore, a suppressive trend in *Crps* expression was evident in DIO mice, supported by a previous study showing intestinal epithelial insulin resistance in DIO mice[29]. These data suggested that insulin signaling upregulated some antimicrobial

peptides (AMPs) in the ileum, which could affect the proportion of *B.massiliensis*.

Other potential promoters of hepatic inflammation such as lipopolysaccharide (LPS) increased in STAM mice (Supplementary Fig. 7a) could be induced by physical barrier dysfunction, or gut leakiness. Along with the comparison between DIO and normal mice, compared with DIO mice, a larger amount of fluorescein isothiocyanate-dextran 4 kDa (FD-4) was observed in STAM mice after its oral gavage, (Supplementary Fig. 7b, c), and it was not reversed by supplementation of insulin (Supplementary Fig. 7d). These data suggest that insulin promotes chemical barrier function, thereby improving gut flora, not physical barrier function.

### Patients with diabetes and NASH showed a similar signature of gut flora to that of STAM mice which was preserved by insulin treatment

To assess whether the dysbiosis observed in STAM mice, presumably due to decreased insulin action in the gut, was also relevant to patients with diabetes and NASH, we performed 16 S metagenomic analyses using fecal DNA samples of patients comorbid with diabetes and NASH. Fecal samples were collected from 27 patients (15 male and 12 female patients) admitted to the Department of Gastroenterology, the University of Tokyo Hospital, and their pathological diagnosis was confirmed by liver biopsy. Of these, 20 patients (13 male and 7 female patients) were diagnosed with both diabetes and NASH, classified into Matteoni class 4 or NASH-LC. Their samples were used for the following analyses (Fig. 5a). Their mean age was 68.0 years old (Fig. 5b), and their mean BMI was 29.4 (Fig. 5c), suggesting the existence of both insulin resistance and insufficient compensation of insulin secretion, similar to the characteristics of STAM mice. Significant differences were not found in the available clinical features (age, sex, BMI, AST, ALT, total cholesterol, triglyceride, and pathological grade) presented in supplementary materials (Supplementary Table 4). Available information of the oral glucose tolerance test (OGTT), plasma deoxycholic acid and cholic acid of the patients were presented in Supplementary Table 5 although we did not perform statistical comparison owing to inadequacy of data.

Although a significant change was not observed in Analysis of Compositions of Microbiomes with Bias Correction 2 (ANCOM-BC2[30], Supplementary Fig. 8a, Supplementary Table 7) and *Bacteroides massiliensis* itself (Supplementary Fig. 8b, c) as well as diversities (Supplementary Fig. 8d–f), the abundance of another genetically similar bacterium, *Bacteroides vulgatus*, was changed significantly (Fig. 5d, unpaired t test). The bacterium was focused on because we could refer to the previous reports, which suggest the relevance between the abundance of genus Bacteroides and insulin-use[31], or show genetic similarity between *Bacteroides vulgatus* and *Bacteroides massiliensis*[32]. In addition, the abundance of *Bacteroides vulgatus* was also known to increase in the gut flora of fatty liver patients[33, 34]. Host preference is also assumed between mice and humans.

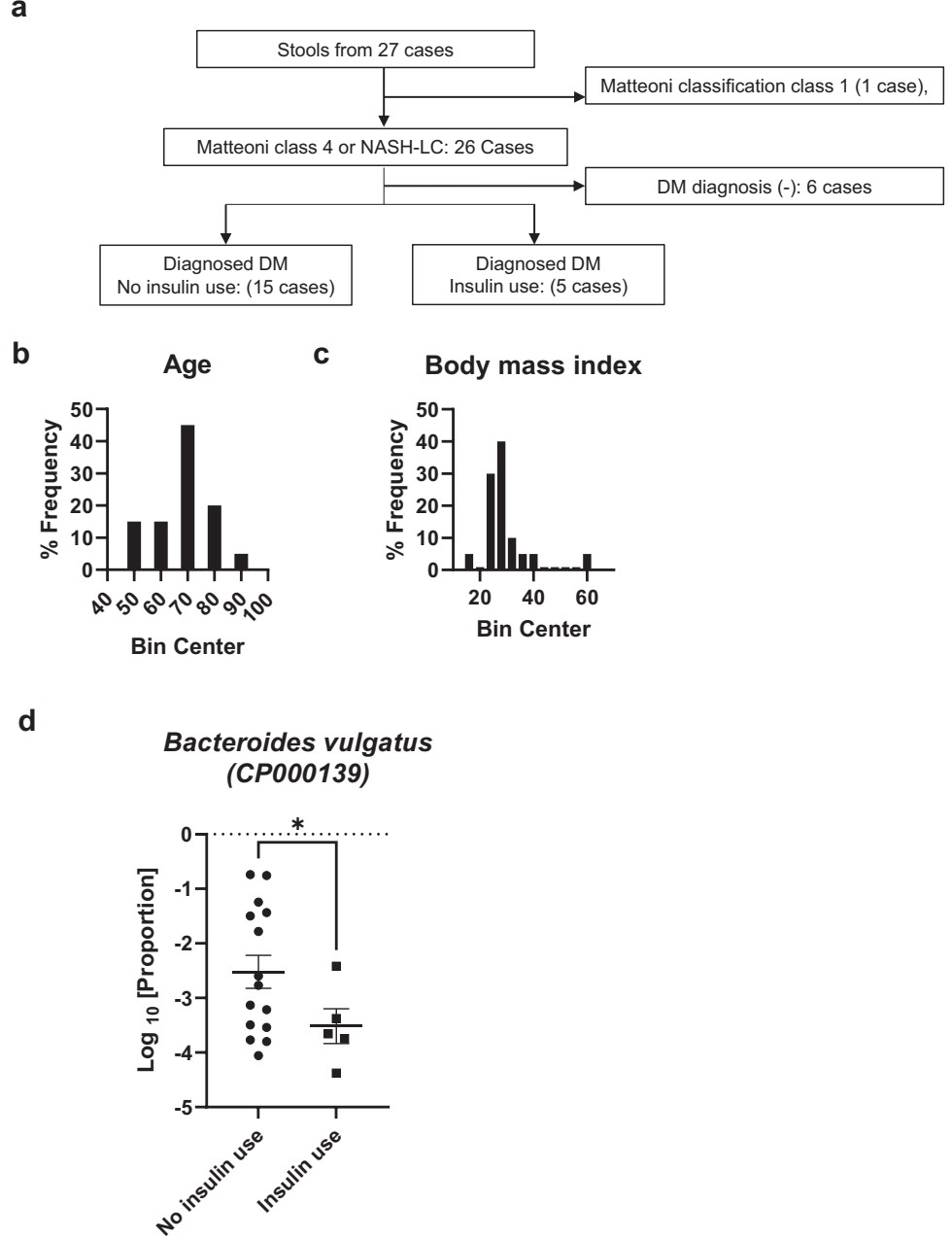

**Fig. 5 | 16S metagenomic signature alteration by insulin use in NASH patients with diabetes.** The schematic protocol for participant selection for assessment of human diabetic NASH patients' gut microbiota (**a**). Distribution of all participant ages (**b**) and body mass index (BMI, **c**). Logarithm 10 of proportion to all reads of *Bacteroides vulgatus* (CP000139) (**d**) in non-insulin-treated (*no* insulin use, $n = 15$) and insulin-treated NASH patients (insulin use, $n = 5$). Values of the data are expressed as mean ± SEM (**d**). *$P < 0.05$, 2-sided unpaired t test with Welch's correction (**d**). Source data are provided as a Source Data file. The exact $P$ values are provided in Supplementary Data 3 unless they are below 0.0001.

## Antibiotic treatment or fecal microbiota transplantation suppressed the development of HCC in STAM mice

To evaluate the impact of dysbiosis on the development of HCC in STAM mice, STAM mice were orally administrated triple antibiotics, including aminobenzyl penicillin (ABPC), neomycin (NEO), and vancomycin (VCM), to establish intestinal sterilization (Fig. 6a).

Real-time PCR analysis confirmed the depletion of most bacteria, with a relative abundance of 16 S rDNA being decreased to nearly one-thirtieth of the control (Supplementary Fig. 9a), and 16 S metagenomic analysis confirmed that *B. massiliensis*, a suspected promoter in this pathological etiology, was remarkably suppressed by this triple antibiotic treatment (Fig. 6b) along with its belonging classifications (Bacteroidetes>Bacteroidia>Bacteroidales>Bacteroidaceae>Bacteroides

Supplementary Fig. 9b–f). These changes were accompanied by a significant decrease in blood glucose (Supplementary Fig. 9g) without changes in hepatic accumulation of triglyceride (Supplementary Fig. 9h) and cholesterol (Supplementary Fig. 9i). In the antibiotics-treated group, the number of visible tumors (Fig. 6c, d) and their maximum diameter (Fig. 6e) were significantly suppressed, supporting the contribution of the certain bacteria in the gut flora of STAM mice.

Next, we carried out fecal microbiota transplantation (FMT), considering the difficulty of inoculation because of the strict anaerobe of *B. massiliensis*[32]. FMT from normal mice after intestinal sterilization (Fig. 6f) induced significant suppression of *B. massiliensis* (Fig. 6g), while *Clostridium* cluster XI, a suspected producer of DCA[8], was not suppressed (Supplementary Fig. 10a). The maximum diameter and

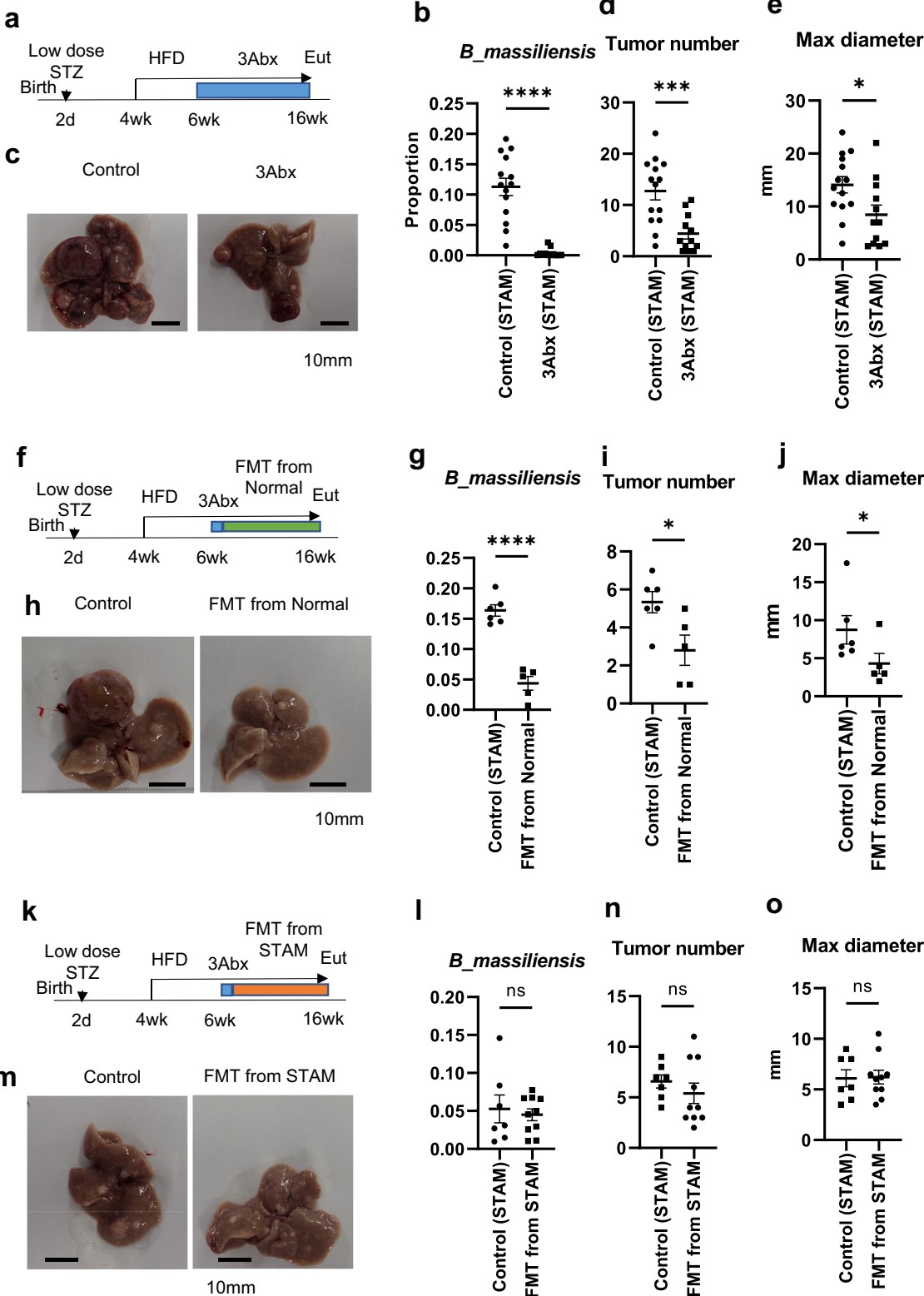

tumor number of individual mice were significantly suppressed (Fig. 6h–j) along with blood glucose level (Supplementary Fig. 10a) by FMT.

Third, FMT from STAM mice to STAM mice (Fig. 6k) did not suppress the proportion of *B. massiliensis* (Fig. 6l), hepatocarcinogenesis (Fig. 6m–o), and blood glucose level (Supplementary Fig. 10b). A significant change of neither triglyceride nor cholesterol accumulation was not observed in these experiments (Supplementary Fig. 10a,

b). These data suggest that certain bacteria, usually suppressed in normal flora, promoted the development of tumor in the liver in STAM mice.

Additionally, to elucidate the role of bile salt hydrolase (BSH) in developing liver tumors in STAM mice, we administrated BSH inhibitor to STAM mice to inhibit deconjugation of secondary bile acids (Supplementary Fig. 10c). After administrating the compound (GR-7; Gut restricted-7[35]), we observed a reduction in the number of liver tumors

**Fig. 6 | Suppression of hepatocarcinogenesis by continuous treatment with triple antibiotics or fecal microbiota transplantation (FMT) in STAM mice.** The experiment protocol for triple antibiotic treatment (ampicillin, neomycin, vancomycin; 3Abx) of STAM mice (**a**). Read proportion of *Bacteroides massiliensis* (**b**), representative macroscopic appearance of the liver (**c**), visible tumor number (**d**) and maximum diameter (**e**) of individual mouse in each group. Control (STAM) *n* = 14, 3Abx (STAM) *n* = 12, *P < 0.05, ***P < 0.001, ****P < 0.0001, with 2-sided unpaired t test (**b, d, e**). Scale bar, 10 mm for macroscopic images. The experiment protocol for fecal microbiota transplantation (FMT) from normal mice (**f**). Read proportion of *Bacteroides massiliensis* (**g**), representative macroscopic appearance of the liver (**h**), visible tumor number (**i**) and maximum diameter (**j**) of individual mouse in each group. Control (STAM) *n* = 6, FMT from Normal *n* = 5, *P < 0.05, ****P < 0.0001, with 2-sided unpaired t test (**g, i, j**). Scale bar, 10 mm for macroscopic images. The experiment protocol for FMT from STAM mice (**k**). Read proportion of *Bacteroides massiliensis* (**l**), representative macroscopic appearance of the liver (**m**), visible tumor number (**n**) and maximum diameter (**o**) of individual mouse in each group. Control (STAM) *n* = 7, FMT from STAM *n* = 10, 2-sided unpaired t test (**l, n, o**). Scale bar, 10 mm for macroscopic images. Values of the data are expressed as mean ± SEM (**b, d, e, g, i, j, l, n, o**). Source data are provided as a Source Data file. The exact *P* values are provided in Supplementary Data 3 unless they are below 0.0001.

in STAM mice with a trend (*P* = 0.05, Supplementary Fig. 10d), and a significant reduction of *Bacteroides massiliensis* (Supplementary Fig. 10e). In this experiment, it was necessary to use cyclodexitrin as a vehicle, which might lead to increase in *Akkermansia muciniphila* (Supplementary Fig. 10f). Since the increase in this species was previously reported to improve hepatic steatosis[36], the vehicle treatment might have a beneficial effect thereby diminishing the difference by the treatment.

These data support the possibility that certain bacteria including *B. massiliensis* promoted liver tumor development in this model, although gnotobiotic experiments are needed to demonstrate specific mechanisms.

## Intestinal epithelium-specific insulin receptor knockout resulted in impaired intestinal barrier function

The results described above led to the hypothesis that intestinal insulin signaling plays a pivotal role in suppressing dysbiosis and hepatocarcinogenesis in STAM mice. To test this hypothesis, intestinal epithelial-specific insulin receptor-deficient mice (Villin-Cre; IR-flox, ieIRKO mice) were generated to analyze their barrier functions when treated with insulin (Fig. 7a). It was confirmed that the level of expression of insulin receptor (*InsR*) mRNA was remarkably suppressed in ieIRKO-STAM mice compared to Villin-Cre-/-; IR-floxed mice (Supplementary Fig. 11a).

Insulin treatment significantly improved blood glucose levels (Supplementary Fig. 11b), increased body weight (Supplementary Fig. 11c), and epididymal adipose tissue weight (Supplementary Fig. 11d) in control STAM mice and ieIRKO-STAM mice.

In ileum of STAM mice, microscopic analysis revealed fewer eosinophilic granules and decreased alucian blue positive granules in crypt, which were ameliorated by insulin treatment in STAM mice, but not in ieIRKO-STAM mice (Fig. 7b, c). These microscopic findings in crypt granules, which represented antimicrobial peptides (AMPs) in Paneth cells, suggested that ieIR signaling deficiency resulted in dysfunctional AMPs production in Paneth cells.

Functional annotation analysis [The Database for Annotation, Visualization, and Integrated Discovery (DAVID) v6.8] using the gene list including genes that were up-regulated in insulin-treated STAM mice compared to non-treated STAM mice and insulin-treated ieIRKO-STAM mice (Supplementary Fig. 11e, see **Methods** for details) suggested a strong relationship with AMPs. Real-time PCR analysis also revealed that all genomic subtypes of alpha-defensins (*Defa3, Defa5, Defa20, Defa21, Defa22, Defa23, Defa24, Defa26*) and *Crps* were significantly up-regulated by insulin treatment in STAM mice, while upregulation of these genes in response to insulin was not observed in ieIRKO-STAM mice, different from control STAM mice treated with insulin (Fig. 7d, e, Supplementary Fig. 12b, c). Other well-known barrier factors, *Lyz1, Mmp7, Reg3g, pIgR, Occludin, Zo-1, Muc2* were not significantly upregulated by insulin treatment (Supplementary Fig. 11f). Comparing ieIRKO mice with the littermate control mice under the same conditions (DIO, STAM, insulin-treated STAM, Supplementary Fig. 12a), suppressed expression of AMPs was observed in ieIRKO mice regardless of the feeding and treatment conditions, while Control-DIO

and STAM-Ins, both of which were hyperinsulinemic, showed increased expression (Supplementary Fig. 12b, c).

On the other hand, a trend toward shortening of the colon (*P* = 0.09; Supplementary Fig. 13a) and significantly decreased cecal weight (Supplementary Fig. 13b) were found in insulin-treated ieIRKO-STAM mice compared with insulin-treated control STAM mice, while the knocking out of IR was also confirmed in colon (Supplementary Fig. 13c). The expression profile of the known barrier factors, *Defb1, Defb4, Occludin, Zo-1, Muc2,* and *pIgR* (Supplementary Fig. 13c) in the colon may not be correlated with either insulin signaling or barrier function of the individual types of mice.

To elucidate the direct effect of insulin on the ileum, ex vivo insulin stimulation experiments[37] were performed. Four-hour insulin stimulation led to significant upregulation of *Defa22, Defa20, Defa21* (*P* < 0.05), with a similar trend shown for *Defa3, Defa5,* and *Defa26* (*P* < 0.10), suggesting that alpha-defensins were induced directly by insulin in the ileum (Fig. 7f).

It was reported that the intracellular insulin receptor signaling cascade includes Akt/PKB[29]. TSC2 is one of the main mediators which suppress mTORC1 signaling. Therefore, to assess the role of this signaling pathway downstream of the insulin receptor, ileal gene expression analysis using intestinal epithelial Akt1 and Akt2 double deficient mice (ieAkt1/2DKO) and intestinal epithelial Akt1, Akt2 and TSC2 triple deficient mice (ieAkt1/2TSC2TKO) was conducted. Reductions of phosphorylated Akt (pAkt) and phosphorylated ribosomal protein S6 (pS6), which was a marker of mTORC1 signal, were observed in ieAkt1/2DKO mice (Supplementary Fig. 14a). In ieAkt1/2DKO mice, significant suppression of AMP production was also detected (Supplementary Fig. 14b). In ieAkt1/2TSC2TKO mice, resuscitation of pS6 despite reduction of pAkt was observed (Supplementary Fig. 14c), and resuscitation of AMP production also occurred (Supplementary Fig. 14d), suggesting that Akt-TSC2/mTORC1 signal facilitates AMP production in intestinal epithelium.

## Insulin treatment failed to suppress HCC in ieIRKO-STAM mice

Next, we performed 16S metagenomic analyses. A 16 S metagenomic profiling of non-insulin-treated STAM mice, insulin-treated STAM mice, and insulin-treated ieIRKO-STAM mice strongly suggested that insulin treatment alleviated the dysbiosis induced by intestinal epithelial insulin signal insufficiency. Especially, *Bacteroides massiliensis* was upregulated in STAM mice and was normalized by insulin treatment, while insulin failed to decrease it in ieIRKO-STAM mice, suggesting that the abundance of *Bacteroides massiliensis* is regulated by intestinal insulin signaling (Fig. 8a, Supplementary Fig. 15a). The suppressive effect of insulin on *Bacteroidetes* was significantly canceled in ieIRKO-STAM mice, as was the effect on *Firmicutes* (Supplementary Fig. 15b). Moreover, at the species level, bacteria suppressed by insulin treatment in control STAM mice, proliferated in insulin-treated ieIRKO-STAM mice to the level of non-treated control STAM mice (Supplementary Fig. 15c).

In parallel with insulin treatment-induced changes in the microbiome, insulin treatment significantly reduced the number of tumors in control STAM mice, but failed to do so in ieIRKO-STAM mice

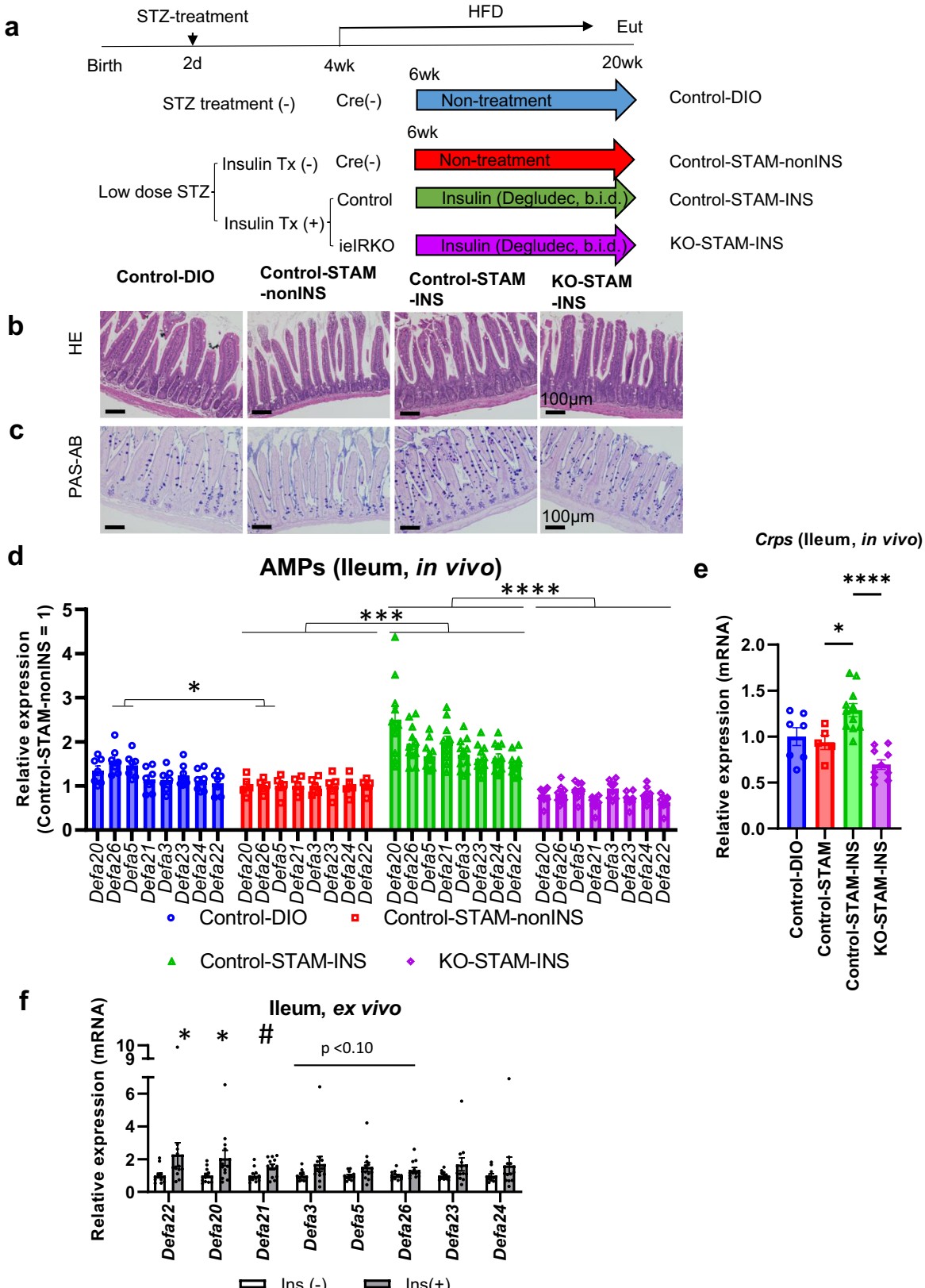

(Fig. 8b–d). The differences in response to insulin between the genotypes were associated with differences in maximum diameter of the tumor, and hepatic accumulation of DCA (Fig. 8e, Supplementary Fig. 15d, 15e).

Both insulin-treated control STAM mice and ieIRKO-STAM mice were comparable in background hepatic triglyceride and cholesterol accumulation (Supplementary Fig. 15f) and liver weight (Supplementary Fig. 15g). Insulin treatment suppressed the level of

**Fig. 7 | Upregulation of AMPs expression via intestinal epithelial insulin receptor in STAM mice.** The experiment protocol for insulin treatment of intestinal epithelial insulin receptor knock out STAM (ieIRKO-STAM) mice (**a**). HE staining (**b**) and PAS-alucian blue staining (**c**) of ileum. Relative expression level of ileal AMPs (**d**) and *Crps* (**e**), determined by qPCR. Control-DIO *n* = 7, non-insulin-treated control STAM mice (Control-STAM-nonINS) *n* = 5, insulin-treated control STAM mice (Control-STAM-INS) *n* = 11, insulin-treated ieIRKO-STAM mice (ieIRKO-STAM-INS) *n* = 10, *P* < 0.05, ***P* < 0.001, ****P* < 0.0001, qPCR analysis with one-way ANOVA,

Šídák's multiple comparisons test between adjacent two groups. Relative expression levels of alpha defensins in ex vivo insulin-stimulated ileal sectional samples. *n* = 12, *P* < 0.05, 2-sided Mann-Whitney's U test for *Defa22, Defa20, Defa3, Defa5, Defa26, Defa23*, and *Defa24*. #*P* < 0.05, 2-sided unpaired t test for *Defa21* (**f**). Values of the data are expressed as mean ± SEM (**d, e, f**). Source data are provided as a Source Data file. The exact *P* values are provided in Supplementary Data 3 unless they are below 0.0001.

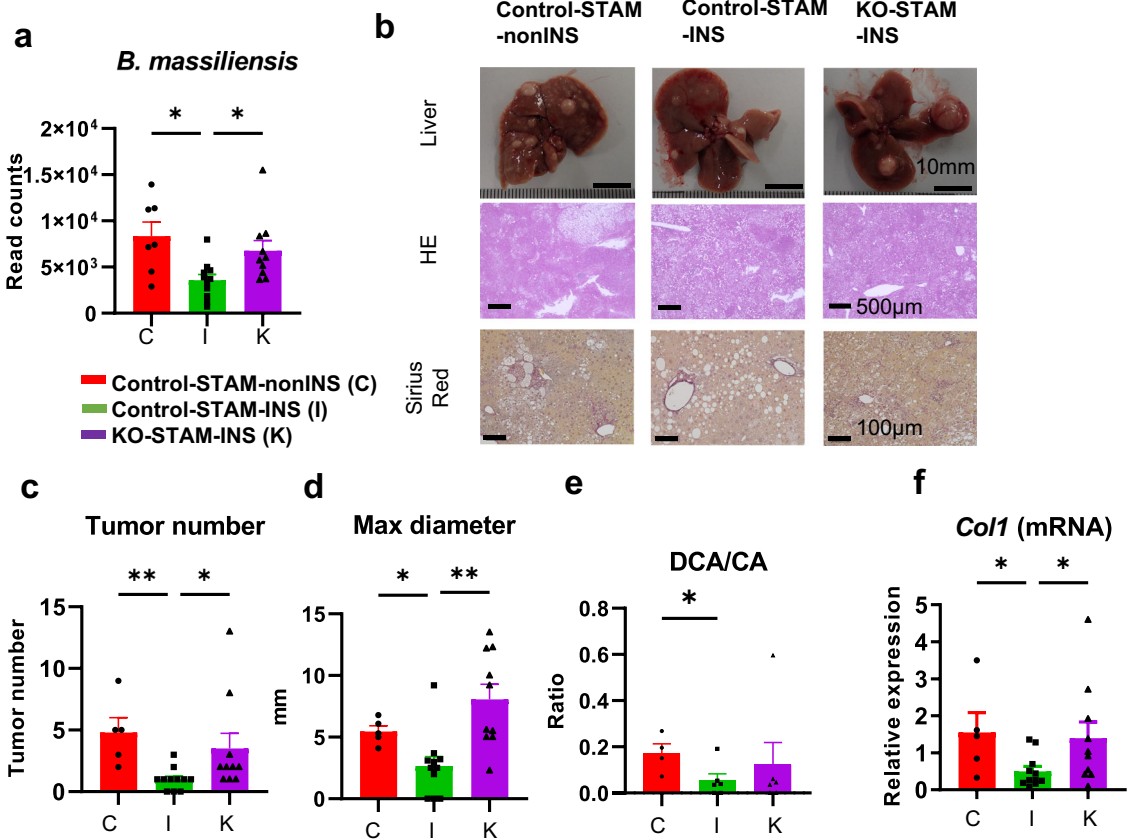

**Fig. 8 | Preservation of dysbiosis and inhibition of hepatocarcinogenesis via intestinal epithelial insulin receptor in STAM mice.** Read count of *Bacteroides massiliensis*, determined by 16 S metagenomic analysis (**a**). non-insulin-treated control-STAM mice (Control-STAM-nonINS) *n* = 7, insulin-treated control STAM mice (Control-STAM-INS) *n* = 11, insulin-treated intestinal epithelial insulin receptor knock out STAM mice (ieIRKO-STAM-INS) *n* = 10. DNA was extracted from collectable fresh fecal samples from each mouse. Macroscopic image and microscopic appearance (HE, Sirius red staining) of representative liver from each group (**b**). Scale bar, 10 mm for macroscopic images and 100 μm for microscopic images.

Tumor number (**c**) and maximum diameter (**d**) of each mouse. Representative deoxycholic acid (DCA) / cholic acid (CA) ratio in liver, determined by LC/MS (**e**). Relative mRNA expression level of *Col1* (**f**). Control-STAM-nonINS *n* = 5, Control-STAM-INS *n* = 11, ieIRKO-STAM-INS *n* = 10, *P* < 0.05, ***P* < 0.01, one-way ANOVA, with multiple comparisons test except for Fig. 8e. In Fig. 8e, Control-STAM-nonINS *n* = 4, Control-STAM-INS *n* = 6, ieIRKO-STAM-INS *n* = 6. *P* < 0.05, 2-sided Mann Whitney's U test. Values of the data are expressed as mean ± SEM (**a, c, d, e, f**). Source data are provided as a Source Data file. The exact *P* values are provided in Supplementary Data 3 unless they are below 0.0001.

expression of *Col1* but failed to do so in ieIRKO mice (Fig. 8f), which was supported by Sirius red staining (Fig. 8b). Liver injury in insulin-treated ieIRKO-STAM mice was confirmed by mRNA expression of *Tnf-α, Ccl2, Il1b* in the liver (*P* = 0.09, *P* < 0.05, *P* < 0.05, respectively, insulin-treated STAM mice vs. insulin-treated ieIRKO-STAM mice; Supplementary Fig. 15h), and upregulated liver 16 S rDNA (not significant but with a mean fold change of 2.3; Supplementary Fig. 15i) also suggest the increased bacterial translocation in insulin-treated ieIRKO-STAM mice.

Although the STAM mouse model has the advantage of being readily available, there is no ruling out the independent carcinogenic effect of STZ, a DNA-damaging alkylating agent, as shown in whole exome sequencing. Therefore, an additional experiment was performed under more physiological conditions, where ieIRKO mice and

control mice were fed with a high-fat diet for 18 months. In this experiment, the maximum diameter of liver tumor (Fig. 9a, c) and plasma alpha-fetoprotein concentration (Fig. 9d) were significantly higher in ieIRKO mice than in control mice, while the number of tumors was similar (Fig. 9b). When these mice were fed with a normal chow diet, few liver tumors were found (Supplementary Fig. 16a, b). Therefore, ieIR signal plays a role against pressure toward hepatocarcinogenesis related to HFD feeding. It was also confirmed that the level of expression of *InsR* (Fig. 9e), *Crps* (Fig. 9f), and *Defa3, Defa21, Defa23*, and *Defa26* (Supplementary Fig. 16c) was significantly suppressed in the ileum of ieIRKO mice in HFD feeding background. A 16 S metagenomic profiling of these mice suggested dysbiosis in ieIRKO mice, where the proportion of phylum *Proteobacteria* was significantly higher than in control mice (Supplementary Fig. 16d).

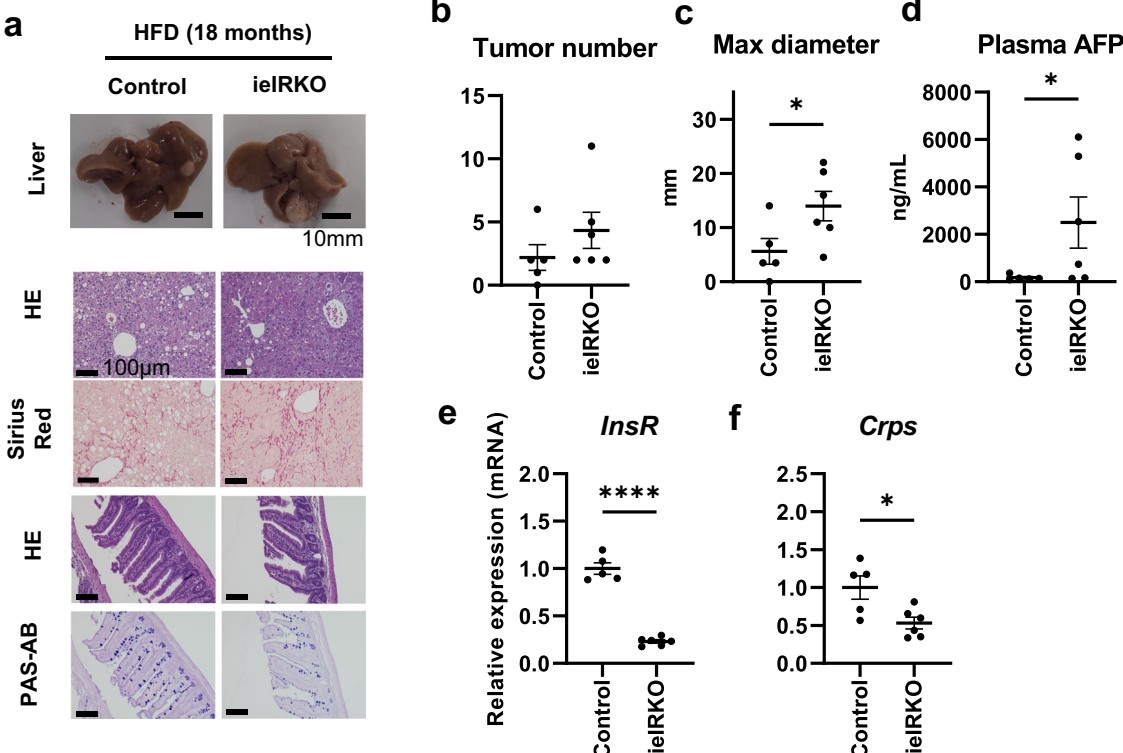

**Fig. 9 | Phenotypes of continuous high-fat-diet-fed ieIRKO mice at 18 months of age.** Representative image of liver and ileum (**a**). Liver sections were stained with HE and Sirius red. Ileal sections were stained with HE and PAS-AB. Scale bar, 10 mm for macroscopic image, 100 μm for microscopic image. Visible tumor number (**b**) and maximum diameter (**c**) in liver of each mouse. Plasma α-fetoprotein (**d**).

Relative mRNA expression of *InsR* (**e**) and *Crps* (**f**). Control ($n = 5$) and ieIRKO ($n = 6$) (**b, c, d, e, f**), *$P < 0.05$, ****$P < 0.0001$, 2-sided Mann Whitney's U test (**b, c, d**), and 2-sided unpaired t test (**e, f**). Values of the data are expressed as mean ± SEM (**b, c, d, e, f**). Source data are provided as a Source Data file. The exact *P* values are provided in Supplementary Data 3 unless they are below 0.0001.

Taken together, these data suggest that insulin action in intestinal epithelial cells plays a protective role against the development of hepatocellular carcinoma associated with diabetes and NASH.

## Discussion

Among the various cancers epidemiologically associated with diabetes, hepatocellular carcinoma is one of the highest relative risks associated with diabetes[1]. On the other hand, recent studies have revealed that a growing number of NAFLD/NASH patients develop HCC[14], while the number of patients with viral hepatitis has been continuously decreasing. Given that diabetes is a known strong risk factor for NAFLD/NASH, management of diabetes may prove vital to suppressing the occurrence of HCC.

While there are many animal models available for NASH-HCC research[38], some exhibit morbid obesity associated with genetic manipulation, and some others are induced by a specific chemical agent or diet, independently of insulin resistance or diabetes. Thus, most models may prove inappropriate for studies aimed at exploring the pathogenesis of NASH/HCC associated with diabetes and developing therapeutic strategies for the disease. Moreover, previous epidemiological studies have shown that Asian subjects with diabetes and NASH exhibit milder obesity than Caucasian subjects[15], and that, presumably due to differences in the pathogenesis of diabetes between the two ethnic groups, Asian subjects have a lower insulin-secretory ability against insulin resistance[4,5].

In this regard, STAM mice, which develop diabetes, NASH, and HCC with an injection of low-dose STZ on birth and continuous high-fat diet (HFD) feeding from weaning[16], may represent an excellent model to explore the mechanism and therapeutic strategy for NASH/HCC associated with diabetes. Indeed, mice treated with STZ and HFD, which show insulin resistance and decompensation of insulin-

secretion have been widely used as a model of type 2 diabetes[39–42]. Moreover, previous studies have demonstrated that the somatic mutation signature in tumors of STAM mice is similar to that of HCC associated with NASH in humans[43]. In this study, we have also confirmed a common somatic mutation signature in both tumor and non-tumor regions, suggesting that these mutations are driver mutations. Therefore, hyperglycemia could be a candidate promoter of HCC with driver mutations in hepatocytes of STAM mice. However, PHZ, an SGLT1/2 inhibitor, which ameliorates hyperglycemia without enhancing insulin secretion or action, fails to suppress HCC in STAM mice, suggesting that amelioration of hyperglycemia per se is insufficient to protect from HCC in STAM mice and in patients with diabetes and NASH. Previous studies showed that the pathological grade of NASH and occurrence of HCC was suppressed by treatment with SGLT2 inhibitors in some morbidly obese rodent models[44,45], suggesting that inhibition of SGLT may be beneficial for the suppression of NASH/NCC associated with severe insulin resistance.

Since diabetes is caused by decreased insulin action in various tissues, impaired insulin action in the liver or other tissues may lead to carcinogenesis supported by driver mutations in STAM mice and in patients with diabetes and NASH. Indeed, different from PHZ treatment, insulin treatment suppresses HCC in STAM mice. In the liver, insulin treatment ameliorates the Warburg effect with the associated reduction of HIF-1α and its downstream molecules. This may contribute to the suppression of tumor growth, but may not be sufficient for preventing tumor development.

Recent studies have shown that HFD feeding induces HCC in mice treated with a chemical agent, which causes somatic mutations in the liver, through the promotion of dysbiosis and production of toxic secondary bile acids[8]. In the current study, we have also found that HFD feeding and STZ injection led to dysbiosis associated with

changes in secondary bile acids and thus dysbiosis might play a causal role in the development of HCC, although it remains undetermined exactly which bacteria play a role in this model. A similar type of dysbiosis is shown in patients with diabetes and NASH, but not in those treated with insulin. These data prompted us to test the hypothesis that insulin action in the gut controls intestinal barrier function that protects against dysbiosis induced by HFD and subsequent development of HCC. Indeed, insulin treatment significantly increases the expression of AMPs in the ileum in vivo and in vitro. Moreover, STAM mice with disruption of the insulin receptor in intestinal epithelial cells (ieIRKO mice) are shown to be resistant to insulin treatment in terms of preserving dysbiosis and suppressing the development of HCC. Furthermore, ieIRKO mice show markedly increased HCC on long-term HFD feeding, suggesting that impaired insulin signaling, such as insulin resistance, in the gut plays a key role in the development of naturally-occurring HCC associated with obesity and an unhealthy diet. Previous studies have shown that insulin resistance is observed in the intestinal epithelium of DIO mice[29] and that expression of AMPs is decreased in the gut of ieIRKO mice[13,46]. Therefore, under diabetic conditions, insulin action in intestinal epithelial cells may be impaired, and patients comorbid with NASH could be susceptible to HCC. Indeed, the current study revealed that supplementation of insulin effect in the gut recovers expression of AMPs and ameliorates dysbiosis through insulin receptor in the gut.

In this study, it remains undetermined that *B. massiliensis* plays a role in promoting HCC in this model. Although isolation and implantation to germ-free mice are needed to assert robust conclusions, it is extremely difficult to isolate this specific bacterium. The vulnerability of germ-free mice and hygienic control against polyurea and high fat diet feeding to generate STAM mice in germ-free isolators are other technical issues to conquer. We could not determine any species with statistical significance in non-biased differential analysis such as ANCOM, and many clinical parameters also remain unknown in the human study. Thus, further efforts using animal models are needed, and the collection of various specimens and clinical data from patients with NASH and diabetes are also required to clarify the direct relationship between the specific species and HCC development in the future.

In conclusion, intestinal insulin signaling plays a protective role against the development of nonalcoholic steatohepatitis and hepatocellular carcinoma associated with diabetes in mice, and treatments leading to the preservation of insulin action in the gut of patients with diabetes could be useful for prevention of NASH and HCC.

## Methods

### Mice

Animal experimental procedures were approved by the Institutional Animal Care and Use Committee of the National Center for Global Health and Medicine (approved protocol number; Med-P16-113, Med-P16-114) and the Animal Care Committee of the University of Tokyo (approved protocol number; 21077, 2022-A044). All relevant ethical guidelines were followed.

The mouse model of NASH (STAM™ mouse) was generated according to the protocol described previously as follows[16]. STAM™ mice was generated by subcutaneous injection of low-dose streptozotocin (STZ, 200 µg/head) to 2-day-old male C57B/6 J mice (CLEA Japan, Tokyo, Japan) and by feeding High Fat Diet 32 (CLEA Japan, consisting of 25.5% protein, 32.0% fat, 29.4% nitrogen-free extract, 4.0% ash, 2.9% fiber, and 6.2% water, HFD) from the age of 4 weeks[16]. All the mice used in this study was male mice because previous study had revealed that female mice treated with STZ and HFD never developed HCC by the absence of NASH-based fibrosis[16].

The experiment involving administration of insulin or phlorizin until the age of 20 weeks (Fig. 2a) was performed on request of the University of Tokyo at Stelic Institute & Co., Inc. and the other experiments were performed at the University of Tokyo and Research Institute of the National Center for Global Health and Medicine. The mice were euthanized when we found severe impairment of activity and decrease of more than 25% of body weight independent of tumor size to save animal welfare.

All mice were housed under a 12-hour light/12-hour dark cycle at macroenvironmental temperature and humidity ranges of 20 to 22 °C and 40 to 60%, respectively, and had free access to sterile water and pellet food. Except for STAM™ mice and diet-induced-obese mice, CE-2 (CLEA Japan, Tokyo, Japan, consisting of 25.05% protein, 4.77% fat, 49.82% nitrogen-free extract, 7.04% ash, 4.44% fiber, and 8.88% water) was given, unless otherwise indicated.

For an intestinal epithelial-specific deletion model, Villin-Cre mice (The Jackson Laboratory, Bar Harbor, US) and insulin receptor-floxed mice[47], Akt1-floxed mice[47], Akt2-floxed mice[47], TSC2-floxed mice[47] were bred to produce Villin-Cre; IR-floxed mice (ieIRKO mice), Villin-Cre; Akt1 Akt2-double floxed mice (ieAkt1/2DKO mice), Villin-Cre; Akt1 Akt2 TSC2-triple floxed mice (ieAkt1/2TSC2TKO mice). Primer sequences for genotyping are shown in Supplementary Data 1 in this article. The ieIRKO mice were treated as above[16] to produce ieIRKO-STAM mice. IeAkt1/2DKO mice and ie Akt1/2TSC2TKO mice were used at the age of 8 weeks (Supplementary Figure 14), and the phenotype of ieIRKO-STAM mice were observed at the age of 20 weeks and 18 months on HFD feeding (Fig. 8).

Insulin or phlorizin (PHZ) treatment was implemented by subcutaneous injection twice daily (b.i.d.) from 6 weeks of age. For injection, insulin glargine (LANTUS™, Sanofi Aventis) was diluted in citrate buffer (100 mM, pH 4.5), and degludec (TRESIBA™, Novo Nordisc) was diluted in normal saline. The injection dose was determined by blood glucose level, and was gradually increased until 12 U/kg BW (insulin glargine for study at 20 weeks of age), 50U/kgBW (insulin glargine for study at 9 weeks of age), 12U/kgBW (insulin degludec for study at 20 weeks of age). PHZ (Sigma) was dissolved in a solution containing 10% ethanol, 15% DMSO, and 75% saline and was injected subcutaneously (0.4 g/kg) twice daily. In the study involving insulin-treated ieIRKO-STAM mice, control mice were treated with normal saline.

To elucidate how these drugs influence the prognosis of these model mice, we planned both long- and short-term studies. The long-term study was implemented until 20 weeks of age when the mice reached the phase where HCC became evident or the hepatic "burn-out" phase. The short-term study was performed until they reached 9 weeks of age, when the mice exhibited steatosis and fibrosis with increased expression level of relevant genes that allowed for appropriate analysis of treatment effects on changes in gene profiling.

In 9 weeks of age, insulin treatment resulted in amelioration of hyperglycemia and hyperinsulinemia (Fig. 2g, h), while PHZ treatment equally ameliorated hyperglycemia without increase in insulin levels (Fig. 2g, h). These patterns of blood glucose and plasma insulin were maintained in the respective treatment groups until 20 weeks of age when HCC became evident (Supplementary Fig. 2a, b).

Antibiotic treatment was performed by adding 1 g/L of ampicillin (Nacalai), 1 g/L of neomycin (WAKO), 0.5 g/L of vancomycin (WAKO) to drinking water, in C57B/6 J mice from 6 weeks of age to 20 weeks of age.

Fecal microbiota transplantation (FMT) was conducted after 5-days treatment of triple antibiotics in C57B/6 J mice. Immediately after quitting antibiotics administration, fresh fecal suspension in normal saline (0.1 g/mL, supernatant obtained by centrifuge at 600 x g for 1 minute) was administrated by oral gavage once a day for the first 2 weeks. In the next 3 weeks, oral gavage was carried out three times a week. In the subsequent period, oral gavage was conducted once a week. FMT was conducted from 6 weeks of age to 20 weeks of age.

In ex vivo primary tissue culture, C57B/6 J mice at the age of 8 weeks were used.

For inhibition of bile acid hydrolase (BSH), GR-7 (Gut restricted −7, #HY-135747, Medchem express) was fed by oral gavage (20.8 mg/kgBW) with vehicle once a day. The vehicle consisted of 10% DMSO and 18% Sulfobutylether-β-Cyclodextrin (SBE-beta-CD, #HY-17031, Medchem express) in normal saline. GR-7 was given mice from the age of 6 weeks to the age of 20 weeks. The mice in control group were given vehicle.

## Patients

The observational cross-sectional study protocol followed the ethical guidelines of the Declaration of Helsinki and was approved by the University of Tokyo Medical Research Center Ethics Committee (Protocol number 3955, date of approval 27th Dec 2012). 16 S metagenomic analysis was performed on stool DNA samples collected from 27 adult patients who experienced liver biopsy according to the following criteria or clinically diagnosed severe non-B non-C cirrhosis at the University of Tokyo Hospital in Tokyo, Japan, from December 27th 2012 to Jun 30th 2021. The recruitment criteria for liver biopsy in patients without HCC were previously described by an associate researcher[48], which could minimize selection bias comparing insulin user and non-insulin user, because the criteria did not include whether the patients receive insulin treatment. Stool samples were immediately frozen in the hospital ward and transferred with refrigerant to the laboratory freezer in a day. They were stored at minus 80 degrees Celsius in the laboratory until DNA extraction. All patients provided written informed consents. After excluding patients who had never been diagnosed DM (6 cases) and patients who were categorized to Matteoni classification class 2 or 1 (1 case), 20 patients with diabetes who were diagnosed NASH as Matteoni classification class 4 or clinically diagnosed liver cirrhosis were analyzed as follows. The 16 S metagenomic data from 5 patients who were under insulin-treatment were compared with those of remaining 15 patients. Clinical characteristics of these patients were indicated in Supplementary Table 4 and 5. Patients were not excluded by use of antibiotics, proton pump inhibitors, probiotics or other confounding agent. We succeed sequence of all the 20 samples which were attempted. Sample size was determined referring previous studies[31,49,50]. Sample size was also estimated by EZR[51] software (version 4.2.2) as 4 and 16 in two groups respectively, provided by the following parameters given by animal study (Fig. 4e), standard deviation 0.0918, the difference with or without treatment 0.154, statistical significance (α) 5%, statistical power (1-β) 80%, the ratio of sample size of two groups 4.

## Tissue sampling

Mice were euthanized by cervical dislocation, and tissues were quickly removed and frozen instantly in liquid nitrogen.

## Gene expression analysis

RNA was extracted using a tissue total RNA mini kit (FAVORGEN). Real-time quantitative PCR was performed using a KAPA SYBR Fast qPCR kit (KAPA BIOSYSTEMS) or a Taqman probe (ABI). Expression levels were normalized to that of the 18 S rRNA gene. Primer sequences are shown in Supplementary Data 1 in this article.

## Whole exome sequence analysis

Whole exome sequence analysis was performed with the cooperation of associate researchers following previous report[52]. Briefly, DNA was extracted from fresh-frozen samples. DNA and RNA were extracted and purified with the Qiagen AllPrep DNA/RNA Mini Kit. Exome capture was performed using SureSelect Mouse All Exon Kit, and sequenced on HiSeq2500. Sequence data were analyzed by karkinos 4.1.11 (modified for mouse genome MGSCv37/mm9). Tumors > 4 mm were excised and subjected to DNA extraction[43].

## DNA Microarray

The specific gene expression patterns in the liver of STAM mice were examined by Agilent Expression Array analysis on request of the researchers by Takara Bio Inc. (Mie, Japan) from total liver RNA with SurePrint G3 Mouse GE microarray 8 x 60K. Total RNA was extracted from snap frozen liver samples from three STAM mice at 20 weeks of age in each group with Qiagen RNeasy Mini Kit. Briefly, according to the manufacturer's information, target samples were prepared with Low Input Quick Amp Labeling Kit, one-color (Agilent). They were hybridized with Gene Expression Hybridization Kit (Agilent) and washed with Gene Expression Wash Buffers Pack (Agilent). GO enrichments were performed using Metascape (version 3.0) and GO TRRUST (version 2). To explore related signaling pathways in the liver of STAM mice, comprehensive hepatic gene expression analyses were performed by microarray between normal mice, non-treated control STAM mice, and insulin-treated STAM mice at HCC phase (20 weeks of age).

## RNA Sequencing analysis

Total RNA extracted from mouse ileal whole wall was treated with Invitrogen™ TURBO™ DNase (Invitrogen), and total RNA sequencing was performed on request of the researchers by Rhelixa, inc. with NovaSeq (Illumina). Fastq files are analyzed with CLC genomics workbench (version 21.0.3, Qiagen) to make alignment referring to mouse genome GRCm38 (mm10). GO enrichments were performed using DAVID v6.8. Highly condensed genes were listed as upregulated by insulin treatment in STAM mice (fold change, >1.5; FDR, $P$ value < 0.05; read number, >4000) and downregulated by intestinal epithelial-specific knockout of insulin receptor (ieIRKO) in insulin-treated STAM mice (fold change, <0.67; FDR, $P$ value, <0.05; read number >4000, Supplementary Fig. 11e).

## Metabolomic analysis

Metabolite extraction and analysis using the CE-TOF/MS and LC-TOF/MS systems were performed on request of the researchers by HMT (Human Metabolome Technologies Inc., Yamagata, Japan), following previous reports[53-55]. Snap-frozen liver samples were prepared from three or four STAM mice in each group at 9 weeks and 20 weeks of age.

For CE-TOF/MS ($n = 3$), approximately 40 mg of frozen liver was immersed into 750 μL of 50% acetonitrile/Milli-Q water containing internal standards at 0 °C to inactivate enzymes. The tissue was then homogenized and centrifuged (2300 × g, 5 min). Subsequently, the upper aqueous layer was centrifugally filtered to remove proteins. The filtrate was vaporized and resuspended in water for capillary electrophoresis coupled to mass spectrometry (CE-MS) analysis.

Cationic compounds were measured in the positive mode of metabolome analysis using Agilent CE TOFMS system (Agilent Technologies), and anionic compounds were measured in the positive and negative modes of metabolome analysis using Agilent CE and Agilent 6460 TripleQuad LC/MS systems (Agilent Technologies) following previous reports[53-55].

For LC-TOF/MS ($n = 4$, for 9 weeks of age, $n = 3$, for 20 weeks of age), approximately 45 mg of frozen liver was immersed into 1% formic acid in acetonitrile containing internal standards and homogenized. The lysate was centrifuged (200 x g, 2 min, 3 times) and supernatants were stored. The pellets were then homogenized again, followed by addition of 167 μL Milli-Q water, and centrifuged. The supernatants were added to the former one. Subsequently, these were centrifugally filtered, and phospholipids removed with the solid phase extraction method. The filtrate was vaporized and resuspended in 100 μL 50% isopropanol (v/v) for LC-TOF/MS

Cationic compounds were measured in the positive mode of metabolome analysis using Agilent 1200 series RRLC system SL

(Agilent Technologies), and anionic compounds were measured in the negative mode of the same system.

In the liver of STAM mice, metabolomics (Liquid Chromatography-Mass spectrometry; LC/MS) also revealed that, of the 98 candidate molecules, 13 were both up-regulated in the STAM model (FC[STAM-NON/Normal]>2, n = 3) and down-regulated by insulin treatment (FC[STAM-INS/STAM-NON] <0.5, n = 3) (Fig. 4a), which included taurochenodeoxycholic acid and taurolithocholic acid, in addition to 11 fatty acids (Fig. 4b).

Metabolome measurements and PCoA analysis were performed at a service facility of Human Metabolome Technologies, Inc. with MasterHands (version 2.16.0.15 for Fig. 4a, b, and Supplementary Figs. 4d, 4e, 6a, 6b version 2.17.1.11 for Supplementary Figs. 4f, 5, developed by Keio University) and SampleStat (version 3.14) respectively.

## 16S metagenomic analysis

16S ribosomal RNA was analyzed using fecal DNA extracted using QIAamp Fast DNA Stool Mini Kit (QIAGEN). Human 16 S metagenomic study was performed in the University of Tokyo Hospital, Department of Diabetes and Metabolic Diseases. Sterilization of associated equipment, devices and bench thoroughly conducted.

The 16 S rRNA sequencing library was constructed according to the Illumina 16 S Metagenomic Sequencing Library Preparation protocol (Illumina) targeting the V3 and V4 hypervariable regions of the 16 S rRNA genes using primers 341 F (5′-CCTACGGGNGGCWGCAG-3′) and 805 R (5′-GACTACHVGGGTATCTAATCC-3′),. KAPA HiFi HotStart Ready Mix (KAPA) was used for the PCRs (95 C, 3 min; 25 cycles of 95 C 30 sec, 55 C, 30 sec, 72 C, 30 sec; 72 C, 5 min; 4 C to hold). The sequence of primers with illumina adaptors were TCGTCGGCAGCGT-CAGATGTGTATAAGAGACAG and GTCTCGTGGGCTCGGA-GATGTGTATAAGAGACAGG. Index PCR was performed with these amplicons with Nextera XT Index Kit v2 Set A (Illumina) to produce 16 S rRNA amplicon library. The proportion of spiked-in PhiX control was 20%.

For 2 × 300 bp paired-end sequencing of the V3 and V4 hypervariable regions on the MiSeq platform, Illumina MiSeq v3 Reagent Kit was used in each experiment. DNA sequences were processed using Miseq Control Software (MCS), and BaseSpace Onsite 16 S Metagenomics App version 1.1.0 (Illumina, Reference; Greengenes). In human study, the average read depth was 67000 reads, and the minimum one was 24000, and the bacterial abundance was expressed as read proportion to all read. Primary component analysis in human study was conducted by Graphpad Prism 9.5.0 software.

Only in the validation of the experiment in which FMT from STAM was conducted, V4 region was amplified, and sequenced in 2 x 150 bp paired-end with iSeq100 system.

*Akkermansia muciniphila(AY271254), Bacteroides massiliensis(AY126616), Prevotellamassilia timonensis(NR144750.1), Acetatifactor muris(HM989805), Bacteroides acidifaciens(ABO21164), Muribaculum intestinale(NR_144616.1)* was selected by following criteria in Supplementary Fig. 15c, ratio of Control-STAM-INS to control-STAM-nonINS was less than 1, and the maximum group mean was more than 5%.

For human microbiome study, the fulfilled STORMS[56] (Strengthening The Organizing and Reporting of Microbiome Studies) checklist is shown as Supplementary Data 2, and species count in csv format generated from BaseSpace 16 S Metagenomics App is shown in Source Data.

## Biochemical assays

Protein samples were obtained using a standard liver buffer. Western blots were performed using Chemi-Lumi One L (Nacalai) or Chemi-Lumi One Super (Nacalai). Primary antibodies, such as Hif1a (1:1000, Novus, #NB100-105), pHSL (1:1000, Cell Signaling Technology, #4139), tHSL (1:1000, Cell Signaling Technology, CST#4107), pS6 (1:2000, Cell Signaling Technology, #5364), tS6 (1:1000, Cell Signaling Technology,

#2217), pAkt (1:2000, Cell Signaling Technology, #4060), and tAkt (1:1000, Cell Signaling Technology, #4691), beta actin (1:12500, Sigma, #A4448) were diluted as indicated above[57]. Secondary HRP-conjugated antibodies (SantaCruz, #sc2357) were diluted at 1:5000, except for anti-mouse IgG (Jackson, #115-035-062, 1:20000). Quantitative densitometric analysis was exerted by gel analyzer in ImageJ software (version 1.52). Triglycerides, cholesterol and NEFA were measured using LabAssay™ Triglyceride and NEFA (FujiFilm, WAKO), respectively. Insulin was measured using a high-sensitive insulin ELISA kit (Morinaga). Plasma AFP was measured using Mouse alpha-Fetoprotein/AFP Quantikine ELISA Kit (R&D SYSTEMS) Plasma LPS was measured using LPS ELISA Kit (CUSABIO)For FD-4 (Fluorescein isothiocyanate–dextran– 4 kDa) assay, mice were fasted 4 hours before oral feeding and for the duration of the experiment. 4 kDa FITC-Dextran (dissolved at 40 mg/mL in PBS, administered at 20uL/gBW) was fed by oral gavage. 4 hours later (no drinking, no feeding), plasma was collected from mice, and fluorescence intensity at Excitation: 490 nm and Emission: 530 nm was read.

## Histological analysis

For the histological examination of mice administered insulin or PHZ to be performed until the age of 20 weeks (Fig. 2a), the liver samples were fixed in OCT compound, and in other experiments, the liver samples were fixed in 4% buffered formalin overnight and embedded in paraffin. All samples were sectioned 5 μm thick and stained with hematoxylin-eosin. Nonalcoholic fatty liver disease (NAFLD) activity was evaluated by the NAFLD activity score. The following was evaluated semiquantitatively: steatosis (0–3 points), lobular inflammation (0–2 points), hepatocellular ballooning (0–2 points), and fibrosis (0–4 points)[58] For fibrosis analysis, liver samples were also stained with sirius red. In the experiment involving insulin or PHZ administration until the age of 20 weeks (Fig. 2a), Sirius red positive area proportion was measured by Image J (version 1.52) in 3 images per mouse on request of the researchers at Stelic Institute & Co., Inc. Intestinal samples were also stained with PAS-alucian blue stain.

## Primary tissue culture

Referring to previous report[37], the mouse at 8 weeks of age ileal tissues were washed with phosphate buffered saline and cut into 1-cm lengths, opened, placed in 24-well plates, and kept in RPMI 1640 with 5% FBS, 7.5% gelatin, D(+)-glucose (final concentration, 200 mM), and antibiotics. Tissues were then cultured with 100 nM of human recombinant insulin (Humarin R, Eli Lilly) at 37 °C, 5% $CO_2$.

## Quantification and statistical analysis

Data are shown as mean ± SEM. The numbers of replicates are expressed as N in the figure legend, which represents the number of mice available for studies in vivo as indicated. Statistical analyses were conducted using Graph Pad Prism 9.5.0 software. Differences between two groups was assessed for significance using an unpaired two-tailed t test or Mann-Whitney's U test. One-way ANOVA and multiple comparison were performed as described in figure legends. The statistically significant values were $P < 0.05$. Sample size estimation was carried out by EZ-R software (Version 4.2.2, Version of Rcmdr was 2.8-0.[51]) By c-Bioinformatics inc. on request, ANCOM-BC2 (ver2.0.2)[30] was conducted in all the aligned species by BaseSpace 16 S Metagenomics App (version 1.1.0).

## Reporting summary

Further information on research design is available in the Nature Portfolio Reporting Summary linked to this article.

# Data availability

The whole exome sequencing and metagenomic data from mice generated in this study have been deposited in the SRA database

under accession code PRJNA866345. The RNA sequencing data from the ileum of mice generated in this study have been deposited in the GEO database under accession code GSE210876. The microarray data from the liver of mice generated in this study have been deposited in the GEO database under accession code GSE210517. The human metagenomic data generated in this study have been deposited in the DDBJ database under study accession code JGAS000574 and dataset code JGAD000700 [https://humandbs.biosciencedbc.jp/en/hum0371-v1]. To obtain data set in DDBJ, data users need to apply an application for Using NBDC Human Data to reach the Controlled-access Data, following the "Ethical Guidelines for Life Science and Medical Research Involving Human Subjects," which is based on Japan's Personal Information Protection Law. How to access is shown in the following URL; https://humandbs.biosciencedbc.jp/en/data-use. Source data are provided with this paper. Mouse genome MGSCv37/mm9 (for Whole exome sequencing), GRCm38/mm10 (for RNA sequencing), Greengenes (for 16 S metagenomics) were used for reference. Source data are provided with this paper.

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

## Acknowledgements

This work has been funded by a grant for the Research Program on Hepatitis from Japan Agency for Medical Research and Development (JP17fk0210304, JP18fk0210040, JP19fk0210040, JP20fk0210040, JP21fk0210090 to K.U.); a Grant-in-Aid for Challenging Exploratory Research (21659227, to K.U.). We thank Dr. C. Ronald Kan (Joslin Diabetes Center) and Dr. Morris Birnbaum (Pfizer) for sharing Akt1 floxed mice and Akt2 floxed mice, respectively, and Mr. Fumiya Takahashi, Ms. Yuko Masaki, Ms. Yasuko Sakuma, Ms. Reiko Homma, Ms. Mizuki Chosa, and Ms. Yasuko Ota for their technical assistance with animal care, Ms. Minori Matsuda for the excellent secretarial work. We thank Ms. Miwa Tamura-Nakano and Ms. Chinatsu Oyama (Communal Laboratory, National Center for Global Health and Medicine) for the assistance of morphological microscopic observation. We thank Dr. Yuki Kawamura (Department of Gastroenterology, Research Institute, National Center for Global Health and Medicine) for her advice on ex vivo ielum experiments.

## Author contributions

K.S. and K.U. designed the study and wrote the manuscript; K.S. and K.U. performed and analyzed most of the experiments; K.E., T.Yamada., T.Nakatsuka., R.T. M.F., and K.K. performed liver biopsies from NASH-suspected patients; M.K., S.Y., K.T., and H.A. acquired and analyzed whole exome sequence data; K.H., K.A., and W.S. helped with metagenomic analyses; T.S., N. Kobayashi, Y.I., M.A., R.B., G.T., N. Kubota, T.Noda., and T.Yamauchi. helped with the experiments; K.U. acquired funding and T. Kadowaki supervised the research.

## Competing interests

The authors declare no competing interests.
