## [Peer Review File · Nature Communications]

Gut insulin action protects from hepatocarcinogenesis in diabetic mice comorbid with nonalcoholic steatohepatitisREVIEWER COMMENTS

Reviewer #1 (Remarks to the Author):

The manuscript by Soeda et al reveals that intestinal insulin signaling protects STAM mice against NASH and HCC. Here the authors treated STAM mice with insulin or Phlorizin to lower hyperglycemia, but only insulin treated STAM mice developed less NASH and HCC. The authors reveal a HIF1 α dependent liver specific mechanism that w/o insulin drives NASH and HCC. Moreover, insulin treated STAM mice showed altered microbiome. authors identified *B. massiliensis* as potential HCC driver, equal to *B. vulgatus* in NASH patient stool. Consistently, when authors inactivated IR in intestine, insulin treatment failed to ameliorate NASH and HCC. Furthermore, long term HFD feeding (w/o STAM) revealed bigger liver tumors in IR IE KO mice indicating that intestinal insulin protects mice from dysbiosis and NASH/HCC.

I have only minor concerns:

The paper is well written, but I want to point to the fact that authors should describe each figure after each other (e.g. Fig1a....Fig. 1B.....Fig1.C). otherwise this confuses the reader. In this context, Fig. 2G and H are only described much later in material and methods section and show similar findings as Fig. S2A and B that were mentioned in the results part.

The Fig 2 has to be separated into two figs. the HIF1 α part deserves its own fig.

in the end, the authors switch from their STAM model to long term HFD feeding to induce HCC. There, the normal chow fed mice are the missing controls. Just in case, the authors feed mice 18 months a HFD- the mice develop liver tumors. what when you feed mice an control diet this time? do they develop liver tumors?

Collectively, these essential points have to be addressed by the authors before publication.

Reviewer #2 (Remarks to the Author):

Soeda et al. have demonstrated the protective role of insulin against hepatocarcinogenesis in diabetic NASH-HCC model mice and patients with diabetes and NASH. The proposed mechanism is that insulin treatment maintains the intestinal barrier function and suppresses dysbiosis of gut microbiota through activating intestinal insulin signaling and stimulating production of antimicrobial peptides. Overall, the findings of the study are novel and interesting. However, the mechanistic story is not clear and strong. The study needs to be significantly strengthened to delineate how specific gut microbial compositions change in response to the progression of NASH-HCC and how insulin signaling facilitates the intestinal production of antimicrobial peptides.

Major comments:

1. Hepatic insulin receptors regulate liver glucose and lipid metabolism and the progression of HCC, which might be the main target of insulin treatment in this study. The authors should provide experimental data to clarify the role of intestinal insulin receptors in the development of HCC as opposed to hepatic insulin receptors, and whether the protective effects of insulin signaling involves both intestinal and hepatic insulin receptors against hepatocarcinogenesis.
2. Please clarify how insulin signaling facilitates the AMP production. Whether endogenous insulin secreted postprandially has any effects on the AMP and gut microbiota in healthy humans?
3. Please show data of the altered tight junction proteins to support the maintenance of epithelial barrier function and integrity with insulin treatment.
4. The authors provided a large amount of data showing the protective effects resulting from insulin treatment, including suppression of hepatocarcinogenesis, preservation of fat-storing function in WAT, amelioration of fibrosis and chronic inflammation, suppression of cell proliferation pathway, et al. These observations should be organized well and support the central hypothesis.
5. The authors did not provide the bile acid data in the clinical samples. This is necessary to support the findings in the animal models.
6. It is necessary to confirm the biotransformation of bile acids by *B. massiliensis* whether through the in vitro experiments or functional analysis of metagenomic data.
7. It is reasonable that antibiotic treatment could relieve HCC, which is consistent to previous report (PMID: 23803760). However, the results are not relevant to this study as the use of broad-spectrum antibiotics suppresses not only key bacteria such as *B. massiliensis*, but also other strains with 7 α -dehydroxylation activity such as *Clostridium*. Germ-free or antibiotic-treated mice with *B. massiliensis* colonization will be more convincing. The authors also need to analyze the bile acid levels in this experiment.
8. The authors showed that gut microbiota-depleted STAM model suppressed the HCC development compared with STAM-control. For the establishment of the causal relationship between the *B. massiliensis* and HCC development, inoculation/transplant of the *B. massiliensis* in a STAM-ABPC model is a necessary step of the mechanistic study.

9. Previous studies have shown that insulin signaling activation stimulates lipid synthesis in liver, which has also been mentioned by the authors (Line 137). Differently, the authors observed the amelioration of lipid accumulation after insulin treatment. Please explain the controversy.

Minor comments:

1. Epididymal white adipose tissue should be replaced with epididymal white adipose tissue in legend of Figure 1D
2. Except for Fasn, the relative expression of ACC (acetyl-CoA carboxylase) should be detected in Figure S3.
3. Line169: epididymal should be corrected to epididymal.
4. The contents of hepatic TC and TG should be detected among the STAM-NON, STAM-INS and STAM-PHZ groups.
5. The relative expression of α -SMA (smooth muscle actin), another hepatic fibrogenic biomarker, should be detected among the STAM-NON, STAM-INS and STAM-PHZ groups.
6. The data of Col1 relative expression between the figure 2F and figure 2I were same ?
7. Please reorganize the Fig 2I.
8. Metabolic flux was necessary to clarify the suppressed Warburg effect by insulin.
9. Which methods have been used to select the differential metabolites, and please offer the results of PCA or PLS-DA.
10. 'TaurochenoDCA' in figure 3B, do you mean 'Taurochenodeoxycholic acid(TCDCA)'?
11. As shown in figure 3B, you observed elevated levels of TCDCA and TCLA in STAM-NON relative to normal. The expression of hepatic bile acid-synthesis related enzymes (CYP7A1, CYP8B1, CYP27A1, CYP7B1) should be detected to see whether the alternative pathway of bile acid biosynthesis has been activated in STAM-NON mice.
12. Please supplement the PCoA plot in Figure S5.
13. Line254: 'and genetically close to *B. massiliensis*' Please add the related reference.
14. Please offer the detailed pathological diagnosis of the 27 patients and other related clinical parameters including OGTT, ITT, ALT, AST, TC and TG.
15. Please offer the relative abundance of *B. massiliensis* in metagenomic data from clinical patients with diabetes and NASH treated with insulin.
16. The hepatic metabolic profiles among the Normal, STAM-control and STAM-ABPC groups should be provided.
17. The phylum and genus level of gut microbiota among the Normal, STAM-control and STAM-ABPC groups should be added.
18. Figure 5C: The ordinate name was wrong. 'STAM-ABPM' should be corrected to 'STAM-ABPC'.
19. The blood glucose, body weight and eWAT were basically the same between the control-STAM-INS and KO-STAM-INS groups, and please explain this phenomenon.
20. KO-STAM-nonINS controls are missing in Fig 6. These are important controls that should be included.
21. What did the sum (species) include? Please list all the species involved.
22. Please provide the bile acid profiles of figure 3B and figure 7E.
23. Were the results of bile acids and gut microbiome from HFD-induced model consistent with the previous results from the STAM model?

Reviewer #3 (Remarks to the Author):

The manuscript by Soeda et al. investigated the insulin action in the HCC development in STAM mouse model. They showed that glucose lowering effect of insulin does not account for its suppressive effect of HCC development in the diabetes-NASH STAM mice, since glucose reabsorption inhibitor did not show such an effect. They further showed that intestinal epithelial IR deletion abolished the insulin effect. In addition, they showed that insulin effect on HCC inhibition in STAM mice is through gut microbiota-mediated barrier function. It is a well-written manuscript and novel in elucidating insulin's action on gut microbiota in diabetes-NASH mice that could develop to HCC. However, this reviewer has several concerns that may need to be addressed to improve the manuscript.

1. Microbiota study is largely descriptive. The authors found changes in gut microbiota, such as *B. massiliensis* and conclude that dysbiosis plays a causal role in the development of HCC. The contribution of those dysregulated bacteria to the HCC development needs to be demonstrated. The authors indicate those bacteria have high BSH activity that can deconjugate bile acids. It is suggested to inhibit BSH activity genetically or pharmacologically and evaluate the effects on HCC development in the STAM model. Such a clear cause-effect study will clearly explain the involvement of gut microbiota in the HCC development.
2. Not clear why some data points involve 10 mice but other only 3 mice. Some data with 3 replicates are not convincing to draw a conclusion.

3. Fig 1 is mainly a characterization of the model. If it has been performed previously, this part should be reduced; if not, several other important parameters, such as dyslipidemia and hepatic steatosis should be displayed in fig 1.
4. Line 112, DIO mice also show increased low-grade inflammation (pro-inflammatory cytokine expression)
5. Page 11, CD63 is not the major fatty acid transport in the liver, other transports, such as FATP2, should be measured.
6. The authors showed that plasma NEFA was suppressed by insulin and attributed it to reduced lipolysis of white adipose tissue. How about brown fat? Brown fat tissues utilize fatty acid that may cause the decreased NEFA in the circulation. In other words, white adipose browning plays a critical role in lipid metabolism, which should be determined.
7. Gut dysbiosis is attributable for barrier dysfunction, but no barrier function was measured. Does STAM mice have elevated serum LPS levels and hepatic bacteria? Does insulin action in the intestine prevent gut leakiness that associated with tight junction proteins?
8. Page 19, if the effects of insulin treatment on adipose tissue lipolysis through intestine insulin receptor, why ielRKO mice had similar eWAT compared to control mice?
9. in other words, how intestinal insulin signaling affecting adipose lipolysis is not clearly demonstrated.
10. Page 21 line 344, $P=0.09$ is not significant

Reviewer #1

*> The manuscript by Soeda et al reveals that intestinal insulin signaling protects STAM mice against NASH and HCC. Here the authors treated STAM mice with insulin or Phlorizin to lower hyperglycemia, but only insulin treated STAM mice developed less NASH and HCC. The authors reveal a HIF1a dependent liver specific mechanism that w/o insulin drives NASH and HCC. Moreover, insulin treated STAM mice showed altered microbiome. Authors identified *B. massiliensis* as potential HCC driver, equal to *B. vulgatus* in NASH patient stool. Consistently, when authors inactivated IR in intestine, insulin treatment failed to ameliorate NASH and HCC. Furthermore, long term HFD feeding (w/o STAM) revealed bigger liver tumors in IR^{IE} KO mice indicating that intestinal insulin protects mice from dysbiosis and NASH/HCC.*

I have only minor concerns:

We are very much grateful to the Reviewer #1 for the quite positive comments. We believe that we have revised the manuscript entirely following the comments by Reviewer #1, as described below.

> The paper is well written, but I want to point to the fact that authors should describe each figure after each other (e.g. Fig 1A...Fig. 1B...Fig 1.C). otherwise this confuses the reader. In this context, Fig. 2G and H are only described much later in material and methods section and show similar findings as Fig. S2A and B that were mentioned in the results part.

We apologize for the inappropriate order of the description in the original manuscript. Following the comment, in the revised version of the manuscript, figures are explained by the order of numbering.

> The Fig 2 has to be separated into two figs. The HIF1a part deserves its own fig.

As suggested by the comment, the Fig. 2 is separated into two parts (Fig.2 and Fig.3) in the revised manuscript.

> in the end, the authors switch from their STAM model to long term HFD feeding to induce HCC. There, the normal chow fed mice are the missing controls. Just in case, the authors feed mice 18 months a HFD- the mice develop liver tumors. What when you feed mice a control diet this time? Do they develop liver tumors?

Thank you very much for the comment. We bred *ieIRKO* mice and littermate floxed mice with

normal chow diet for 18 months and observed no tumor lesion in ieIRKO mice while one out of ten mice burdened one tumor in the floxed mice. As mentioned previously, the data suggested that HFD feeding promoted development of liver tumor. The data are shown in the supplementary figures in the revised manuscript (Fig. S16A, B).

Collectively, these essential points have to be addressed by the authors before publication.
As described above, all the points raised by Reviewer #1 have been addressed and we believe that the manuscript has been appropriately revised.

Reviewer #2 (Remarks to the Author):

>Soeda et al. have demonstrated the protective role of insulin against hepatocarcinogenesis in diabetic NASH-HCC model mice and patients with diabetes and NASH. The proposed mechanism is that insulin treatment maintains the intestinal barrier function and suppresses dysbiosis of gut microbiota through activating intestinal insulin signaling and stimulating production of antimicrobial peptides. Overall, the findings of the study are novel and interesting. However, the mechanistic story is not clear and strong. The study needs to be significantly strengthened to delineate how specific gut microbial compositions change in response to the progression of NASH-HCC and how insulin signaling facilitates the intestinal production of antimicrobial peptides.

We greatly appreciate that Reviewer #2 found our work of interest and novelty, and gave quite useful comments to improve the manuscript.

Major comments:

>1. Hepatic insulin receptors regulate liver glucose and lipid metabolism and the progression of HCC, which might be the main target of insulin treatment in this study. The authors should provide experimental data to clarify the role of intestinal insulin receptors in the development of HCC as opposed to hepatic insulin receptors, and whether the protective effects of insulin signaling involves both intestinal and hepatic insulin receptors against hepatocarcinogenesis.

As suggested by the reviewer #2 and discussed in the manuscript, hepatic insulin signaling appears to play a role in the development of hepatocellular carcinoma. For instance, our colleague (Sakurai, Kubota, Yamauchi, Kadowaki, *et. al.* Sci Rep. 2017; 7: 5387.) revealed that knocking out of insulin substrate -1 (IRS-1) resulted in reduction of hepatocarcinogenesis in model mice treated with DEN in two weeks of age, although these mice show almost normal glucose metabolism, different from our model. We think that insulin treatment to STAM mice may have beneficial effect on suppression of the development of hepatocellular carcinoma at least in part through hepatic insulin signaling as pointed by Reviewer #2. Indeed, insulin treatment improves the Warburg effect in the liver of STAM mice. In this study, however, we wanted to focus on the unraveled function of insulin in gut in terms of protection from the development of hepatocellular carcinoma through control of barrier function. We agree that it is important to elucidate the role of hepatic insulin signaling in the development or protection of hepatocellular carcinoma particularly under the diabetic condition. Since knocking out of insulin receptor results in severe impairment of systemic glucose metabolism

and marked hyperinsulinemia, it would be difficult to analyze the pure effect of ablation of hepatic insulin signaling on the development of hepatocellular carcinoma. In the future study, we would like to analyze Lirko (Liver-specific insulin receptor knock out)-STAM mice with normoglycemia by any means to explore the role of hepatic insulin signaling in the development of hepatocellular carcinoma.

2. Please clarify how insulin signaling facilitates the AMP production. Whether endogenous insulin secreted postprandially has any effects on the AMP and gut microbiota in healthy humans?

Thank you very much for the important comment. As reported previously, intestinal epithelial insulin receptor signaling activates Akt protein kinase (Nature Metabolism | VOL 1 | MARCH 2019 | 371–389). To address the mechanism of insulin signaling-induced AMP production, we prepared intestinal epithelial specific Akt1/2-double deficient mice (ieAktDKO mice; Villin-Cre; Akt1 Akt2 double-floxed, Fig. S14A). In the ileum of the control mice and ieAktDKO mice, we confirmed that phosphorylated Akt and ribosomal protein S6 were suppressed by knocking out of Akt1 and Akt2.

We observed that expression of AMPs was suppressed in ieAktDKO mice (Fig. S14B), as observed in ieIRKO mice, suggesting that Akt downstream for IR promotes expression of AMPs. Moreover, we also confirmed that additional knocking out of a putative downstream mediator TSC2 reversed the expression level of AMPs to the level of control floxed mice (Fig. S14C, D). Since TSC2 is known as a suppressive mediator of Akt-mTORC1 signal, these data suggest that insulin signaling facilitates the AMP production via Akt-mTORC1 signal. As the reviewer #2 pointed out, in healthy humans, whether physiological insulin secretion induces AMPs expression and improvement of dysbiosis is important. Combined with the data that STAM mice show decreased AMP production and insulin treatment preserves the levels of AMP production, these data strongly suggest that endogenous insulin regulates AMP production through insulin receptor/Akt/mTORC1 signaling in gut.

A
ieAkt1/2DKO

B

C
ieAkt1/2TSC2TKO

D

Figure S14

>3. Please show data of the altered tight junction proteins to support the maintenance of epithelial barrier function and integrity with insulin treatment.

Thank you very much for the comment. The similar query was given by Reviewer #3. Real time PCR analysis revealed that the change in expression level of Occludin and Zo-1 were not significantly different between in the presence and absence of insulin receptor in STAM mice treated with insulin (Fig. S13C), suggesting that gut insulin signaling *per se* may not play a major role in the regulation of mechanical barrier function under the diabetic condition. These data were included in the manuscript and discussed.

>4. The authors provided a large amount of data showing the protective effects resulting from insulin treatment, including suppression of hepatocarcinogenesis, preservation of fat-storing function in WAT, amelioration of fibrosis and chronic inflammation, suppression of cell proliferation pathway, et al. These observations should be organized well and support the central hypothesis.

Thank you for the useful comment. In the revised manuscript, we did best to organize description to well explain our central hypothesis, that intestinal insulin signaling protect the model mice from dysbiosis and hepatocarcinogenesis.

>5. The authors did not provide the bile acid data in the clinical samples. This is necessary to support the findings in the animal models.

Thank you very much for the comment. The data of bile acids which could be available were shown in the revised manuscript (Supplementary Table 3), although the information was quite limited. In future, we would like to perform a larger study to more profoundly analyze the profiles of bile acids in NASH patients with or without insulin treatment.

>6. It is necessary to confirm the biotransformation of bile acids by *B. massiliensis* whether through the *in vitro* experiments or functional analysis of metagenomic data.

Thank you very much for the useful comment. In fact, *B. massiliensis* is known as strictly anaerobic bacteria (Int J Syst Evol Microbiol. 2005 May;55(Pt 3):1335-1337. doi: 10.1099/ijs.0.63350-0.), which makes its isolation, proliferation, and inoculation very difficult. Instead, we can refer to the database (PATRIC 3.6.12, *Bacteroides massiliensis* B84634 =

Timone 84634 = DSM 17679 = JCM 13223), indicating that the bacteria have bile acid hydrolase (BSH) which is responsible for the first step to synthesis of secondary bile acids.

>7. *It is reasonable that antibiotic treatment could relieve HCC, which is consistent to previous report (PMID: 23803760). However, the results are not relevant to this study as the use of broad-spectrum antibiotics suppresses not only key bacteria such as B. massiliensis, but also other strains with 7 α -dehydroxylation activity such as Clostridium. Germ-free or antibiotic-treated mice with B. massiliensis colonization will be more convincing. The authors also need to analyze the bile acid levels in this experiment.*

Thank you very much for the important comment. Since as mentioned above, it is very difficult to isolate *B. massiliensis*, we performed the fecal microbiota transplantation (FMT) from normal mice to STAM mice. First, intestinal sterilization with ampicillin, neomycin, vancomycin (Fig. 6A) succeeded in remarkable decrease of microbiota in gut (Fig. S9A). Real-time PCR revealed that the treatment resulted in the robust reduction in the 16S rDNA abundance to one thirtieth of that from equal amount of DNA extracted from control stool (Fig. S9A), showing that much more profound suppression of microbiota was achieved compared with ABPC treatment. This intestinal sterilization led to significant suppression of hepatocarcinogenesis in STAM mice (Fig. 6C-E), again suggesting that some bacteria promoted hepatocarcinogenesis in this model mice. Second, FMT from normal mice was carried out to STAM mice after intestinal sterilization (Fig. 6F). Significant suppression of *B. massiliensis* was confirmed by 16S metagenomics, compared with non-treated STAM mice (Fig. 6G). Maximum diameter (Fig. 6I) and tumor number (Fig. 6J) of individual mice was significantly suppressed by FMT. The abundance of genus *Clostridium XI* was not suppressed in this experiment (Fig. S10A), which was the suspected bacteria which are responsible for hepatocarcinogenesis in previous study (Nature. 2013 Jul 4;499(7456):97-101. Obesity-induced gut microbial metabolite promotes liver cancer through senescence secretome). Third, FMT from STAM mice was carried out to STAM mice after intestinal sterilization (Fig. 6K-O). FMT from STAM mice did not suppress hepatocarcinogenesis (Fig. 6N, O), and the proportion of *B. massiliensis* did not change (Fig. 6L) compared with donor STAM mice. Considering that the significant proportional change of the species other than *B. massiliensis* was hardly found, *B. massiliensis* is likely to be responsible for hepatocarcinogenesis in this model.

Fig 6

>8. The authors showed that gut microbiota-depleted STAM model suppressed the HCC development compared with STAM-control. For the establishment of the causal relationship between the *B. massiliensis* and HCC development, inoculation/transplant of the *B. massiliensis* in a STAM-ABPC model is a necessary step of the mechanistic study.

Thank you very much for the comment. As mentioned above, inoculation/transplant of the bacteria was supposed to be very difficult owing to its characteristics which hardly permits exposure to open air. Therefore, instead we performed the FMT experiment as described above (Fig. 6F-J).

>9. Previous studies have shown that insulin signaling activation stimulates lipid synthesis in liver, which has also been mentioned by the authors (Line 137). Differently, the authors observed the amelioration of lipid accumulation after insulin treatment. Please explain the controversy.

Thank you very much for the comment. As pointed by reviewer #2 and described in our manuscript, insulin signaling in hepatocyte promotes the expression of fatty acid synthase (FAS), SREBP, and so on. However, more than 60% of hepatic lipid is originated from plasma NEFA (Donnelly KL, et al. Sources of fatty acids stored in liver and secreted via lipoproteins in patients with nonalcoholic fatty liver disease. *J. Clin. Invest.* 2005;115:1343–1351.), and in diabetes increased lipolysis by decreased insulin action in adipose tissue has more impact on the development of hepatic lipid accumulation rather than increased de novo lipid synthesis in liver. Indeed, pHSL was increased in STAM mice and suppressed by insulin treatment shown in Fig. S3H in the revised manuscript.

Minor comments:

>1. *Epididymal white adipose tissue should be replaced with epididymal white adipose tissue in legend of Figure 1D*

Thank you very much for pointing out. We corrected the typo in the manuscript.

>2. *Except for Fasn, the relative expression of ACC (acetyl-CoA carboxylase) should be detected in Figure S3.*

Thank you very much for the comment. The expression level of ACC was assessed. The similar expression profile with *Fasn* was observed (Fig. S3E). The data are shown in revised manuscript.

>3. Line169: epididymal should be corrected to epididymal.

Thank you very much for pointing out. We corrected the typo in the revised manuscript.

>4. The contents of hepatic TC and TG should be detected among the STAM-NON, STAM-INS and STAM-PHZ groups.

Thank you very much for the comment. Hepatic triglyceride was significantly suppressed by insulin-treatment (Fig. S3C). The reduction was also observed in PHZ-treatment. It was also observed the same trend in hepatic cholesterol, which did not reach statistical significance (Fig. S3D). These results suggest both treatments improve hepatic steatosis in this model mice.

>5. The relative expression of α -SMA (smooth muscle actin), another hepatic fibrogenic biomarker, should be detected among the STAM-NON, STAM-INS and STAM-PHZ groups.

Thank you very much for the comment. In fact, as previously reported, there were very wide dispersion in real time PCR analysis for cytokine level and fibrogenic biomarker in late phase of this model mice (Med Mol Morphol. 2013 Sep;46(3):141-52. doi: 10.1007/s00795-013-0016-1. Epub 2013 Feb 22. A murine model of non-alcoholic steatohepatitis showing evidence of association between diabetes and hepatocellular carcinoma Masato Fujii, Hiroyuki Yoneyama, et al). Although alpha-SMA was analyzed in STAM mice, significant increment

was not observed. Therefore, *Col-1* is selected as a marker of fibrogenesis in this study.

>6. *The data of Col1 relative expression between the figure 2F and figure 2I were same ?*

Thank you very much for the comment. In Fig. 2F, we analyzed the expression level in the late phase in 20 weeks of age. In Fig. 2I, we analyzed in the early phase in 9 weeks of age.

>7. *Please reorganize the Fig 2I.*

Thank you very much for the comment. The expression level of each gene was placed side by side (STAM-NON, STAM-INS, STAM-PHZ).

>8. *Metabolic flux was necessary to clarify the suppressed Warburg effect by insulin.*

Thank you very much for the comment. The issue pointed is very important for specific increment by insulin signaling. To elucidate this issue, to incubate lipid-storing hepatocyte as itself is very important because hepatic steatosis is one of the putative causes to promote expression of *Hif1a* (J Cell Mol Med, 15 (2011), pp. 1329-1338). However, it is very difficult to take primary hepatocyte from model mice that exhibit hepatic steatosis. The *in vivo* study with marker molecules is desired in the future.

>9. *Which methods have been used to select the differential metabolites, and please offer the results of PCA or PLS-DA.*

Thank you very much for the comment. First, to select differential metabolites, we checked up the abundance of each metabolite and fold change with p value by Welch's t test. PCA plot is added to the revised manuscript (Fig. S5).

>10. *'TaurochenoDCA' in figure 3B, do you mean 'Taurochenodeoxycholic acid(TCDCA)'?*

Yes. the word was corrected. Additionally, we apologize for misclassifying the molecule. In fact, taurochenodeoxycholic acid is a taurine-conjugated primary bile acid, which is corrected in the revised manuscript.

>11. *As shown in figure 3B, you observed elevated levels of TCDCA and TCLA in STAM-NON relative to normal. The expression of hepatic bile acid-synthesis related enzymes (CYP7A1,*

CYP8B1, CYP27A1, CYP7B1) should be detected to see whether the alternative pathway of bile acid biosynthesis has been activated in STAM-NON mice.

Thank you very much for the comment. The hepatic expression level of *CYP7A1, CYP8B1, CYP27A1, CYP7B1* were analyzed. Real-time PCR analysis revealed that significant reduction of *Cyp7b1* and *Cyp27a1* were observed in STAM mice compared with DIO mice (Fig. S3J), suggesting that the main cause of increase in secondary bile acids was not induction of these genes.

>12. Please supplement the PCoA plot in Figure S5.

Thank you very much for the comment. The PCoA plot was supplemented in the revised manuscript (Fig. S6B).

>13. Line254: 'and genetically close to *B. massiliensis*' Please add the related reference.

Thank you very much for the comment. The following reference is added to the revised manuscript.

Arch Microbiol. 2010 Jun;192(6):427-35.

14. Please offer the detailed pathological diagnosis of the 27 patients and other related clinical parameters including OGTT, ITT, ALT, AST, TC and TG.

Thank you very much for the comment. Available data were presented in the revised manuscript (Supplementary Table 2 and Table 3). Insulin tolerance test was not conducted in this observational clinical study because the insulin resistance is not essential to diagnose diabetes and NASH. The data were obtained under treatment selected by attending physician depending on the patient's diabetic condition. Patients whose ID was #4 and #7 exhibited impaired glucose tolerance in oral glucose tolerance test, but chronic hyperglycemia was not proved by blood glucose level and HbA1c. Therefore, they were not diagnosed as diabetes. These data and discussion are inserted in revised manuscript.

15. Please offer the relative abundance of B. massiliensis in metagenomic data from clinical patients with diabetes and NASH treated with insulin.

Thank you very much for the comment. The abundance of *B. massiliensis* was assessed by the 16S metagenomic analysis (Fig. S8C). Significant difference between the groups was not observed, which could be caused by difference of host species. As described in the manuscript, genetically close species were suspected to play a role in NASH-DM patients (Fig. 5E). The data and discussion are shown in the revised manuscript.

16. The hepatic metabolic profiles among the Normal, STAM-control and STAM-ABPC groups should be provided.

Thank you very much for the comment. In the revised manuscript, the data about hepatic accumulation of triglyceride and cholesterol were added in the section of antibiotic treatment (Fig. S9H, I).

17. The phylum and genus level of gut microbiota among the Normal, STAM-control and STAM-ABPC groups should be added.

Thank you very much for the comment. For intestinal sterilization, triple antibiotic intervention was conducted in the revised manuscript (Fig. 6A). Each level of classification is presented in the revised manuscript (Fig. S9B-F).

18. Figure 5C: The ordinate name was wrong. 'STAM-ABPM' should be corrected to 'STAM-ABPC'.

Thank you very much for the comment. For intestinal sterilization, triple antibiotic

intervention was conducted in the revised manuscript (Fig. 6A). We corrected the typo in the revised manuscript.

>19. *The blood glucose, body weight and eWAT were basically the same between the control-STAM-INS and KO-STAM-INS groups, and please explain this phenomenon.*

Thank you very much for the comment. The observed data suggest intestinal insulin signaling did not significantly influence obesity and insulin resistance represented by blood glucose, body weight and eWAT weight in this model (Fig. S11B-D).

>20. *KO-STAM-nonINS controls are missing in Fig 6. These are important controls that should be included.*

Thank you very much for the comment. The gene expression data of intestine from KO-STAM-nonINS are shown in the revised manuscript (Fig. S12B, C). Comparing expression of each AMPs of ieIRKO mice with littermate control under the same condition (DIO, STAM, insulin-treated STAM), a trend toward suppression was more evident in insulin-treated STAM mice than other group pairs. The statistical comparison of *Crps* was conducted in Control-DIO, Control STAM-nonINS, KO-STAM-nonINS as in Fig. 7D, Fig. 7E. These data are consistent with the hypothesis that these genes were regulated by ieIR signal.

>21. *What did the sum (species) include? Please list all the species involved.*

Thank you very much for the comment. It included the sum of the abundance of following species. These are clarified in the revised manuscript.

Akkermansia muciniphila(AY271254), *Bacteroides massiliensis*(AY126616), *Prevotellamassilia timonensis*(NR144750.1), *Acetatifactor muris*(HM989805), *Bacteroides acidifaciens*(AB021164), *Muribaculum intestinale*(NR_144616.1).

>22. *Please provide the bile acid profiles of figure 3B and figure 7E.*

Thank you very much for the comment. The abundance of each target was indicated in supplementary table 1 and the data of cholic acid and deoxycholic acid were indicated in Figure S15 respectively.

>23. *Were the results of bile acids and gut microbiome from HFD-induced model consistent*

with the previous results from the STAM model?

Thank you very much for the comment. Previously, it is suggested that 7 α -dehydroxylation activity such as *Clostridium* was responsible for HFD feeding-related promotion of hepatocarcinogenesis by production of secondary bile acids, as reviewer #2 also mentioned above. However, bile salt hydrolase (BSH), which is responsible for upstream metabolic reaction by gut bacteria, is also another important step to produce secondary bile acids as shown in some previous reports. In STAM model, *B.massiliensis* occupied a remarkably large proportion in gut flora, and it was suggested that the flora of STAM mice promoted hepatocarcinogenesis. *B. mass* has genes which express BSH. Taken together, although it is true that *Chlostridium* and 7 α -dehydroxylation are important in hepatocarcinogenesis in the mice on HFD with chemical tumor inducer treatment, under diabetic condition, such as STAM mice, upregulation of *B.massiliensis* or BSH have a pivotal role in hepatocarcinogenesis in response to suppressed AMPs production owing to insufficiency of IR signaling.

Reviewer #3 (Remarks to the Author):

>The manuscript by Soeda et al. investigated the insulin action in the HCC development in STAM mouse model. They showed that glucose lowering effect of insulin does not account for its suppressive effect of HCC development in the diabetes-NASH STAM mice, since glucose reabsorption inhibitor did not show such an effect. They further showed that intestinal epithelial IR deletion abolished the insulin effect. I addition, they showed that insulin effect on HCC inhibition in STAM mice is through gut microbiota-mediated barrier function. It is a well- written manuscript and novel in elucidating insulin's action on gut microbiota in diabetes-NASH mice that could develop to HCC. However, this reviewer has several concerns that may need to be addressed to improve the manuscript.

We are very much grateful to Reviewer #3 for the useful comments that helped to make our manuscript so much improved. As shown below, we have revised the manuscript following the individual comments.

*1. Microbiota study is largely descriptive. The authors found changes in gut microbiota, such as *B. massiliensis* and conclude that dysbiosis plays acausal role in the development of HCC. The contribution of those dysregulated bacteria to the HCC development needs to be*

demonstrated. The authors indicate those bacteria have high BSH activity that can deconjugate bile acids. It is suggested to inhibit BSH activity genetically or pharmacologically and evaluate the effects on HCC development in the STAM model. Such a clear cause-effect study will clearly explain the involvement of gut microbiota in the HCC development.

Thank you very much for the important and useful comment. To elucidate the role of bacterial BSH in developing liver tumor in STAM mice, we conducted repetitive administration of BSH inhibitor to STAM mice. When administrating the compound (Fig. S10C, Gut restricted-7; GR-7, Nat Chem Biol. 2020 Mar; 16(3): 318–326.) which was previously demonstrated to inhibit BSH activity *in vivo*, we found a reduction in the number of liver tumor in STAM mice with a trend ($P = 0.05$), (Fig. S10D) and a significant reduction of the abundance of *Bacteroides massiliensis* (Fig. S10E). Bile acid inhibitor AAA10 was also shown to improve fatty liver after 8 days-administration in rats fed a choline-deficient, high-fat diet (CDAHFD, Inhibition of microbial deconjugation of micellar bile acids protects against intestinal permeability and liver injury. (Biorxiv doi: <https://doi.org/10.1101/2021.03.24.436896>), which is also a model of NASH, although they did not investigate hepatocarcinogenesis in their study. To carry out repetitive administration of GR-7 which is hardly soluble to water, DMSO and beta cyclodextrin were added to the vehicle. It was previously reported that the cyclodextrin improves dysbiosis and steatosis in mouse model (Beneficial Effects of Three Dietary Cyclodextrins on Preventing Fat Accumulation and Remodeling Gut Microbiota in ... Foods 2022, 11, 1118. <https://doi.org/10.3390/foods11081118>), which is known to be hydrolyzed by enzymes produced by intestinal flora and metabolized into short-chain fatty acids (SCFAs). We also observed remarkable increase of abundance in *Akkermansia muciniphila* (Fig. S10F) in vehicle group, which could improve fatty liver. (Appl Environ Microbiol . 2020 Mar 18;86(7):e03004-19. doi: 10.1128/AEM.03004-19. Print 2020 Mar 18. *Akkermansia muciniphila* Prevents Fatty Liver Disease, Decreases Serum Triglycerides, and Maintains Gut Homeostasis) Long term administration (10 weeks) was needed in this study until hepatocarcinogenesis, which also makes it difficult to see difference between vehicle group and GR-7 group because of the beneficial effect of cyclodextrin in vehicle group. This point was different from previous study using CDAHFD rat (doi: <https://doi.org/10.1101/2021.03.24.436896>, see above) in which administration was continued for 8 days. Another important point is that BSH inhibitor was invented through modifications of bile acids, which was known as a strong surfactant to cause barrier dysfunction (Nat Chem Biol. 2020 Mar; 16(3): 318–326.). Improvement of the compound is now on-going (A Gut-Restricted Lithocholic Acid Analog as an Inhibitor of Gut Bacterial Bile Salt Hydrolases ACS Chem. Biol. 2021, 16, 8, 1401–1412). Therefore, at this stage, the only

available and most safe compound to perform administration for long period to mice was GR-7. These data and discussion are shown in revised manuscript.

>2. Not clear why some data points involve 10 mice but other only 3 mice. Some data with 3 replicates are not convincing to draw a conclusion.

Thank you very much for the comment. In fact, mass analysis for multiple target (~300 molecules) was conducted to narrow down the candidate molecules including secondary bile acids (Fig. 4B). It suggested intestinal dysbiosis in STAM mice, which was confirmed by metagenomic analysis for all the mice in the experiment. The linkage between intestinal insulin signal, dysbiosis and secondary bile acids was also observed in the following experiment of insulin-treated ieIRKO-STAM mice (Fig. 8A, E). Multiple experiments conducting mass analysis strongly suggest the linkage between them.

>3. Fig 1 is mainly a characterization of the model. If it has been performed previously, this part should be reduced; if not, several other important parameters, such as dyslipidemia and hepatic steatosis should be displayed in fig 1.

Thank you very much for the comment. In this study, we revealed in STAM model that insufficient insulin signaling in intestine resulted in promotion of hepatocarcinogenesis under the condition of HFD-feeding. Therefore, we selected figures comparing blood glucose,

plasma insulin, number of visible tumors between STAM mice, DIO mice, and normal mice as the beginning of the revised manuscript (Fig.1).

>4. Line 112, DIO mice also show increased low-grade inflammation (pro-inflammatory cytokine expression)

Thank you very much for the comment. As pointed out by the reviewer, we stated this more precisely in the revised version as follows.

Before revision;

In relation to hepatic fibrosis and hepatocarcinogenesis, the expression of hepatic inflammatory cytokines (i.e., TNF- α , 111 CCL2, and IL-6) was elevated in STAM mice (Fig. 1F), unlike in DIO mice.

After revision;

In relation to hepatic fibrosis and hepatocarcinogenesis, the expression of hepatic inflammatory cytokines (i.e., TNF- α , 111 CCL2, and IL-6) was elevated in STAM mice (Fig. 1F), compared with DIO mice.

>5. Page 11, CD63 is not the major fatty acid transport in the liver, other transports, such as FATP2, should be measured.

Thank you very much for the comment. The expression level of *FATP2* was compared along with *CD36*. *FATP2* was significantly induced in STAM mice compared with DIO mice, and both insulin-treatment and PHZ-treatment suppressed its expression. It was suggested that *FATP2* might play a role in improve steatosis in both treatments. These results were shown

(Fig. S3J)

in the revised manuscript (Fig. S3J).

>6. The authors showed that plasma NEFA was suppressed by insulin and attributed it to reduced lipolysis of white adipose tissue. How about brown fat? Brown fat tissues utilize fatty acid that may cause the decreased NEFA in the circulation. In other words, white adipose browning plays a critical role in lipid metabolism, which should be determined.

Thank you very much for the comment. In epididymal white adipose tissue (eWAT), the expression of *UCP1* was induced in STAM mice compared with DIO mice while the both insulin-treatment and PHZ-treatment significantly suppressed the induction (Fig. S3I). It was possible that UCP1 played a role in induction of lipolysis in STAM mice compared with DIO mice. Thus, white adipose browning did not explain the difference in the levels of NEFA among STAM mice, STAM+insulin and STAM+PHZ. These data and discussion were shown in revised manuscript.

>7. Gut dysbiosis is attributable for barrier dysfunction, but no barrier function was measured. Does STAM mice have elevated serum LPS levels and hepatic bacteria? Does insulin action in the intestine prevent gut leakiness that associated with tight junction proteins?

Thank you very much for the comment. Additional experiment revealed that elevated abundance of LPS in plasma in STAM mice compared with DIO mice (Fig. S7A). Elevated abundance of bacterial DNA in liver was observed as in Fig. S15I (16 fold significant increase, Control STAM without insulin treatment vs Control DIO).

To elucidate gut leakiness in STAM mice, FD-4 (Fluorescein isothiocyanate–Carboxymethyl–Dextran – 4kDa) assay was conducted. Compared with DIO mice, a larger amount of FD-4 was observed after its oral gavage (Fig. S7B, C), which was not reversed by insulin supplementation (Fig. S7D). As is shown in Fig. S13C in the revised manuscript (Fig. S8A in the original manuscript), real time PCR analysis revealed that the expression level of *Occludin* and *Zo-1* in colon did not change significantly by knocking out insulin receptor in intestine unlike AMPs, suggesting that the expression of these factors was not regulated by insulin receptor signal. These data and discussion were included in the revised manuscript.

>8. Page 19, if the effects of insulin treatment on adipose tissue lipolysis through intestine insulin receptor, why *ieIRKO* mice had similar *eWAT* compared to control mice?

Thank you very much for the comment. We apologize for the description in the original version of the manuscript that might have led to misunderstanding about our thought. We did not claim that intestinal insulin action suppresses lipolysis in adipose tissue. In the revised manuscript, we clearly stated this point.

Before revision;

Intestinal epithelium-specific insulin receptor knockout resulted in impaired intestinal barrier function

The results described above led to the hypothesis that intestinal insulin signaling plays a pivotal role in suppressing dysbiosis and hepatocarcinogenesis in STAM mice.

After revision;

Intestinal epithelium-specific insulin receptor knockout resulted in impaired intestinal barrier function

The results described above led to the hypothesis that intestinal insulin signaling plays a pivotal role in suppressing dysbiosis and hepatocarcinogenesis in STAM mice, independently of insulin-mediated metabolic functions in other tissues.

>9. in other words, how intestinal insulin signaling affecting adipose lipolysis is not clearly demonstrated.

As stated above, we do not claim that lipolysis is through intestinal insulin action.

>10. Page 21 line 344, $P=0.09$ is not significant

Thank you for the comment. The manuscript was revised as below.

Before revision;

On the other hand, significant shortening of colon ($P = 0.09$; **Fig. S8A**)

After revision;

On the other hand, a trend toward shortening of colon ($P = 0.09$; **Fig. S8A**)

REVIEWER COMMENTS

Reviewer #1 (Remarks to the Author):

The authors performed well in the revision - all my concerns were sufficiently addressed and incorporated. I suggest publication.

Reviewer #2 (Remarks to the Author):

I have carefully reviewed the responses to reviewers' questions as well as the revised manuscript. The authors have addressed most (more than half) of the comments/suggestions I provided. I understand that some of the experiments that I suggested are difficult to do, or time-consuming, but these experiments are necessary and the results to be generated are important to support the proposed mechanisms and the conclusions.

The question 5 asked previously (The authors did not provide the bile acid data in the clinical samples. This is necessary to support the findings in the animal models) was not addressed.

The question 6 (It is necessary to confirm the biotransformation of bile acids by *B. massiliensis* whether through the in vitro experiments or functional analysis of metagenomic data) was partially addressed, but that is not a solid answer.

The question 7 (Germ-free or antibiotic-treated mice with *B. massiliensis* colonization will be more convincing. The authors also need to analyze the bile acid levels in this experiment) was not addressed.

The question 8 was partially addressed.

Minor comments question-8 (Metabolic flux was necessary to clarify the suppressed Warburg effect by insulin) was not addressed.

Minor - 15 (Please offer the relative abundance of *B. massiliensis* in metagenomic data from clinical patients with diabetes and NASH treated with insulin), the answer from the authors was - significant difference between the groups was not observed.

My view of this manuscript remains the same - it is an interesting and novel study. The authors have made significant improvement in the resubmission. However, some key experiments were not performed, and critical data regarding the bile acid metabolic phenotype of the patients and the role of *B. massiliensis* (mechanistic or associative) in the development of NASH and HCC were still missing. Therefore, I'm not convinced by what has been described in the results and conclusion of the paper. I think that the authors should either manage to obtain those critical data by performing additional experiments, or carefully re-write the results, discussion, and conclusion to tone down the mechanistic claims that are not well-supported/justified.

Reviewer #3 (Remarks to the Author):

The authors have addressed my concerns with clarification in the main text and, importantly, with additional major experiments.
I believe it is now publishable.

Reviewer #4 (Remarks to the Author):

Soeda and coauthors evaluated specific mice models of NASH and hepatocellular carcinoma in association with the gut microbiota. Some of the alterations were reversed after using the insulin analogue glargine.

I have found several limitations in this manuscript:

*Conceptual. The authors made a lot of effort to assign STAM mice to the clinical situation of NASH in subjects with diabetes but without obesity. In fact, what they are inducing is insulin deficiency with streptozotocin, and then

they are replacing insulin deficiency with exogenous insulin. The comparable clinical situation would be patients with type 1 diabetes and NASH.

*The subjects evaluated in the clinical study (diabetes and NASH) are probably insulin resistant, not insulin-deficient.

*The statistical power of this particular study in human subjects is very low given the sample size and the several confounding factors that are not even mentioned.

*The microbiota analyses performed in mice and humans would need to be considerably improved taken into account the compositional nature of the multiple comparisons made.

*For instance, the term "dysbiosis" is under extensive controversy. What would be dysbiosis here ?

* The results of the fecal microbiota transplantation performed were essentially the same compared with Antibiotic treatment alone given that implantation of exogenous microbiota (precisely the Bacteroides anaerobic bacteria) was mainly unexplored. A more extensive germ-free model would help here in providing more evidences.

*Identical arguments could be given in the microbiota alterations found in the intestinal insulin receptor knockout mice.

*Insulin Glargine has been described to change endogenous insulin receptors and IGF-I in the liver or improve several NAFLD phenotypes in humans. This was not mentioned in the manuscript.

Reviewer #1 (Remarks to the Author):

The authors performed well in the revision - all my concerns were sufficiently addressed and incooperated. I suggest publication.

We are grateful to your favorable comment. Thank you.

Reviewer #2 (Remarks to the Author):

I have carefully reviewed the responses to reviewers' questions as well as the revised manuscript. The authors have addressed most (more than half) of the comments/suggestions I provided. I understand that some of the experiments that I suggested are difficult to do, or time-consuming, but these experiments are necessary and the results to be generated are important to support the proposed mechanisms and the conclusions.

We appreciate for careful reading of reviewer #2.

The question 5 asked previously (The authors did not provide the bile acid data in the clinical samples. This is necessary to support the findings in the animal models) was not addressed.

Thank you for the comment. We apologize for incapability to supply sufficient data set in human study. However, we could suggest the relationships between metagenomic signature of the patients and animal model. We would like to perform a clinical study to confirm our hypothesis in the near future.

We revised manuscript to suggest uncertainty in this point as follows. (Discussion)

“In this study, it remains undetermined that *B. massiliensis* plays a role in promoting HCC in this model. Although isolation and implantation to germ-free mice are needed to assert robust conclusions, it is extremely difficult to isolate this specific bacterium. Thus, further efforts using animal models are needed, and collection of various specimens and clinical data from patients with NASH and diabetes are also required to clarify the direct relationship between the specific species and HCC development.”

*The question 6 (It is necessary to confirm the biotransformation of bile acids by *B. massiliensis* whether through the *in vitro* experiments or functional analysis of metagenomic data) was partially addressed, but that is not a solid answer.*

Thank you for the useful comment. We presented the possibilities that *B. mass* had bile acid hydrolase activity and increased secondary bile acid from bacterial data base. We would like to try to perform *in vitro* experiment to make hypotheses robust in the future. But at this stage, the following words (underlined) were added to the manuscript.

“The bacterium was reported to have bile acid hydrolase (BSH) responsible for deconjugation of taurine- or glycin-conjugated primary bile acids, based on the database (PATRIC 3.6.12, *Bacteroides massiliensis* B84634 = Timone 84634 = DSM 17679 = JCM 13223), and thus

could be involved in supplying a substrate for secondary bile acids production, although it requires *in vitro* confirmation.

”

*The question 7 (Germ-free or antibiotic-treated mice with *B. massiliensis* colonization will be more convincing. The authors also need to analyze the bile acid levels in this experiment) was not addressed.*

Thank you for useful comment. Because of difficulty to isolate and multiply *B. mass* to carry out experiment for *in vivo* hepatic tumor formation, we performed intestinal sterilization and fecal microbiota transplantation from normal mice or from STAM mice as negative control. The data from these experiments suggest our hypothesis closely because other bacteria hardly explain these phenomena from 16S-metagenomic analysis.

Taken together, we added following sentences in “Discussion” to avoid misleading in this manuscript.

“In this study, it remains undetermined that *B. massiliensis* plays a role in promoting HCC in this model. Although isolation and implantation to germ-free mice are needed to assert robust conclusions, it is extremely difficult to isolate this specific bacterium. Thus, further efforts using animal models are needed, and collection of various specimens and clinical data from patients with NASH and diabetes are also required to clarify the direct relationship between the specific species and HCC development.”

The question 8 was partially addressed.

Thank you for the useful comment. The question was that “>8. *The authors showed that gut microbiota-depleted STAM model suppressed the HCC development compared with STAM-control. For the establishment of the causal relationship between the *B. massiliensis* and HCC development, inoculation/transplant of the *B. massiliensis* in a STAM-ABPC model is a necessary step of the mechanistic study.*” As reviewer #2 pointed out, the mechanism in this model mouse is not demonstrated our hypothesis very strictly. However, the data set in this study suggests that any bacteria including *B. mass* in the flora of STAM mice promoted tumor development and that resident bacteria in normal intestinal environment did not. We scrutinized the text in our manuscript and fixed expressions especially in explanations about mechanisms as above.

Minor comments question-8 (Metabolic flux was necessary to clarify the suppressed Warburg

effect by insulin) was not addressed.

Thank you for the useful comment. Although to demonstrate the role of hepatic insulin signal directly to metabolic flux was very difficult, it was sufficiently suggested that systemic supplementation of insulin signal leads to improvement of hepatic Warburg effect.

Minor - 15 (Please offer the relative abundance of B. massiliensis in metagenomic data from clinical patients with diabetes and NASH treated with insulin), the answer from the authors was - significant difference between the groups was not observed.

Thank you for the comment. From our data, it was not evident insulin treatment could suppress identical bacteria in human and mice. However, relationships were found between insulin treatment and the genetically close bacteria in this study. Among them, *B. vulgatus* was reported to have relationships with human NASH patients (Cell Metab, 25 (2017), pp. 1054-1062, Gastroenterology. 2019 Oct;157(4):1109-1122.) as explained in the manuscript, which appeals importance of this study.

My view of this manuscript remains the same - it is an interesting and novel study. The authors have made significant improvement in the resubmission. However, some key experiments were not performed, and critical data regarding the bile acid metabolic phenotype of the patients and the role of B. massiliensis (mechanistic or associative) in the development of NASH and HCC were still missing. Therefore, I'm not convinced by what has been described in the results and conclusion of the paper. I think that the authors should either manage to obtain those critical data by performing additional experiments, or carefully re-write the results, discussion, and conclusion to tone down the mechanistic claims that are not well-supported/justified.

Thank you for careful reading. We are grateful for positive evaluation of novelty and improvement in the manuscript. We could demonstrate the protective role of gut insulin signaling in the development of NASH and HCC, but it needs much investigation to conclude whether *B. mass* contributes to the mechanism or not. Therefore, we decided to revise again throughout the manuscript carefully to tone down the mechanistic claims appropriately to avoid misleading. Main points of revision were as follows.

In "Introduction", we revised the following sentence as underlined from "via". "Here, we demonstrate the protective role of insulin signaling in gut against hepatocarcinogenesis associated with the maintenance of intestinal barrier function and suppression of dysbiosis

using diabetic NASH-HCC model mice and samples from patients with diabetes and NASH.”

In “Results”, “Antibiotic treatment or fecal microbiota transplantation suppressed the development of HCC in STAM mice” section, we revised and re-wrote careful interpretation.

In “Discussion”, we revised the manuscript and added underlined words and phrases.

“In the current study, we have also found that high-fat feeding and STZ injection led to dysbiosis associated with changes in secondary bile acids and thus dysbiosis might play a causal role in the development of HCC although it remains undetermined that the specific bacteria plays the role in this model .”

“In this study, it remains undetermined that *B. massiliensis* plays a role in promoting HCC in this model. Although isolation and implantation to germ-free mice are needed to assert robust conclusions, it is extremely difficult to isolate this specific bacterium. Thus, further efforts using animal models are needed, and collection of various specimens and clinical data from patients with NASH and diabetes are also required to clarify the direct relationship between the specific species and HCC development.

”

Reviewer #3 (Remarks to the Author):

The authors have addressed my concerns with clarification in the main text and, importantly, with additional major experiments.

I believe it is now publishable.

We are grateful to your favorable comment. Thank you.

Reviewer #4

**Conceptual. The authors made a lot of effort to assign STAM mice to the clinical situation of NASH in subjects with diabetes but without obesity. In fact, what they are inducing is insulin deficiency with streptozotocin, and then they are replacing insulin deficiency with exogenous insulin. The comparable clinical situation would be patients with type 1 diabetes and NASH.*

Thank you very much for the useful comment. The insulin level in this model mice is much lower than diet-induced-obese (DIO) mice, which could give readers impression that it might be insulin-deficient model.

Although administration of low dose streptozotocin reduces insulin secretory capacity, it did not completely suppress insulin secretion. Mice treated with low dose streptozotocin on high fat diet have been widely used as Asian type of type 2 diabetes¹⁻⁴. Indeed, plasma insulin level of STAM mice was maintained in the level of normal-chow-fed mice as was indicated in the manuscript (**Fig.1B, C**). It clearly suggests systemic insulin resistance in this model mice based on the evidence that the glucose level of STAM mice was much higher than normal mice although the plasma insulin level was as high as normal mice. These are mentioned in “Discussion” in the latest version of the manuscript.

It was also mentioned in the manuscript that Asian NASH patients comorbid with diabetes present less obese compared with Caucasian counterpart, as was the case with STAM mice. Taken together, it was reasonable that STAM mice were regarded as a model of NASH with DM in Asia.

1 Take, K. et al. Pharmacological Inhibition of Monoacylglycerol O-Acyltransferase 2 Improves Hyperlipidemia, Obesity, and Diabetes by Change in Intestinal Fat Utilization. PLoS One 11, e0150976, doi:10.1371/journal.pone.0150976 (2016).

2 Eckhardt, B. A. et al. Accelerated osteocyte senescence and skeletal fragility in mice with type 2 diabetes. JCI Insight 5, doi:10.1172/jci.insight.135236 (2020).

3 Kleinert, M. et al. Animal models of obesity and diabetes mellitus. Nat Rev Endocrinol 14, 140-162, doi:10.1038/nrendo.2017.161 (2018).

4 Gilbert, E. R., Fu, Z. & Liu, D. Development of a nongenetic mouse model of type 2 diabetes. Exp Diabetes Res 2011, 416254, doi:10.1155/2011/416254 (2011).

Fig.1

**The subjects evaluated in the clinical study (diabetes and NASH) are probably insulin resistant, not insulin-deficient.*

Thank you very much for the comment. As is pointed out, it is true that the patients in this study presented insulin resistant. However, as is indicated in the manuscript, BMI of the patients is around 25~30 (Fig.5C) which suggest mild obesity and insufficient compensation of insulin secretion. Simultaneous manifestation of insulin resistance, suppression of insulin-secretion and suppression of BMI-increase suggests consistency between the animal model and the population in this clinical study.

It was added to the main text of “Results”, in the section titled by “*Patients with diabetes and NASH showed a similar signature of gut flora to that of STAM mice which was preserved by insulin treatment*” that these patients are probably insulin resistant (“Their mean age was 68.0 years old (Fig. 5B), and their mean BMI was 29.4 (Fig. 5C), suggesting the existence of both insulin resistance and insufficient compensation of insulin secretion, similar to the characteristics of STAM mice.”).

Fig. 5

**The statistical power of this particular study in human subjects is very low given the sample*

size and the several confounding factors that are not even mentioned.

Thank you very much for the comment. As reviewer #4 pointed out, the larger study is warranted to confirm our hypothesis robustly.

Sample size in human study was estimated by animal study shown in **Fig. 4E**, in which STAM mice were treated with insulin analogue. The difference between those with and without treatment was 0.154. The standard deviation of these samples was 0.0918. Therefore, the effect size (Cohen's *d*) was calculated as 1.68 with these data. Given the three parameters as follows; two-tailed, alpha (significance) = 5%, beta (power) = 20%, ratio of sample size of each group was 4, the necessary sample size was calculated as 4 and 16, respectively in each group by EZR software (Version of Rcmdr was 2.8-0. Reference; Bone Marrow Transplantation 2013:48,452-458). These processes were mentioned in the section of "Patients" in "Method".

The estimated sample size was unexpectedly small. This is probably because of the effect size (Cohen's *d*) calculated by animal study. However, we found significance in genetically close bacteria as *B. mass.*

The potential confounding factors checked in this study is shown in supplemental material (**Supplementary table 4, 5**) and addressed in the main text clearly, and compared statistically when the missing value was no more than two. There were no significant confounding factors among these parameters. These data are mentioned in the manuscript.

Difference 0.154, Standard deviation 0.0918; Calculated effect size (Cohen's *d*) 1.68

Fig4

ID	DM diagnosis/insulin treatment	Matteoni classification	Sex	Age	BMI [cm/m ²]	AST [IU/L]	ALT [IU/L]	TC [mg/dL]	TG [mg/dL]
1	DM/Ins(-)	NASH_LC	2	51	29.7	47	64	233	137
2	DM/Ins(-)	NASH_LC	2	75	37.1	24	19	142	85
3	DM/Ins(-)	NASH_LC	1	80	27.8	21	15	175	102
4	nonDM		4	2	84	22.7	34	19	152
5	DM/Ins(-)		4	1	79	26.5	36	29	171
6	DM/Ins(-)		4	2	78	25.8	134	108	223
7	nonDM	NASH_LC	2	71	27.7	51	33	175	57
8	DM/Ins(-)	NASH_LC	2	80	26.3	36	34	173	101
9	DM/Ins(-)		4	1	62	28.7	47	65	232
10	nonDM		4	1	80	25.1	48	49	154
11	DM/Ins(+)		4	1	54	28.8	193	226	170
12	DM/Ins(+)	NASH_LC	4	73	33.3	38	30	166	140

Supplementary table 4

ID	OGTT_0min	OGTT_30min	OGTT_60min	OGTT_120min	Cholic acid [μM]	Deoxycholic acid[μM]
1	60	N.D.	N.D.	N.D.	0	0.5
2	114	N.D.	N.D.	N.D.	0	0
3	99	204	243	265	0	0
4	105	198	260	202	N.D.	N.D.
5	102	208	289	301	1.3	0
6	121	237	311	346	0.4	0
7	78	157	214	237	8.6	3
8	N.D.	N.D.	N.D.	N.D.	0	0.8
9	139	223	298	366	N.D.	N.D.
10	91	178	142	155	N.D.	N.D.
11	N.D.	N.D.	N.D.	N.D.	N.D.	N.D.
12	137	198	206	174	N.D.	N.D.

Supplementary table 5

	Insulin treatment	Age	BMI [cm/m ²]	AST [IU/L]	ALT [IU/L]	TC [mg/dL]	TG [mg/dL]
Average	No insulin use	73.13333	28.67333	40.14286	36.78571	179.4615	117.8462
	Insulin use	65.6	31.56	61.8	61.6	157.2	95.6
Standard Error	No insulin use	2.922235	1.507277	7.952416	7.576965	10.12876	13.64777
	Insulin use	4.132191	8.164068	36.7141	46.05242	6.318623	21.27675
N	No insulin use	15	15	14	14	13	13
	Insulin use	5	5	5	5	5	5
p value		0.17772	0.549502	0.353287	0.360329	0.191748	0.373174

Pathological Stage (Matteoni 4 or NASH LC); P = 0.6126 (Fisher's exact test)

Sex; P = 0.6126 (Fisher's exact test)

Supplementary table 6 (New)

**The microbiota analyses performed in mice and humans would need to be considerably improved taken into account the compositional nature of the multiple comparisons made.*

Thank you very much for the useful comment. As reviewer #4 pointed out, multiple comparison test could lead type I error in omics analyses.

We featured *B. mass* as an indicator of dysbiosis in this model because primary component analysis (PCoA) suggested plasma insulin level is related to primary component 2, to which *B. mass* is a major contributor. Along with PCoA, to find out specific bacteria which relate to pathophysiology and treatment in this model, we listed up fold change and *P* value between each group respectively, where the false discovery needed to be considered.

To avoid type I error, two-stage step-up (Benjamini, Krieger, and Yekutieli, FDR = 0.10, by Graph pad Prism version 9) analysis was added to the data sets in **Fig4C**, **Fig4D**, and the results were shown in **supplementary table 2 and 3** additionally. As was indicated in these tables, the candidate which increased in STAM mice and suppressed by insulin-treatment significantly were limited to *B. mass* and *B. sartorii*, and *B. mass* was the only bacteria whose induction was repeatedly confirmed in the following experiments. Therefore, we focused on the bacteria in this study.

Similarly, the table including q value as a complement for **Fig. 5D** (human study) is also shown in supplementary table 7. From the data, single bacteria could not suggest relationship to insulin-use. Therefore, we combined genetically similar bacteria (*B. vulgatus*, *B. thetaiotaomicron*, *B. caccae*) to *B. mass* and suggested relationship as the first revised manuscript.

	Discovery?	P value	Mean of Normal	Mean of STAM-NON	Difference	SE of difference	t ratio	df	q value
Q_sinus	No	0.466375	0.01708	0.01164	0.005445	0.007069	0.7702	7	0.18655
P_distasonis	Yes	0.028559	0.002377	0.01344	-0.01106	0.004024	2.749	7	0.017951
B_hanseni	No	0.862087	0.01431	0.01583	-0.001517	0.008416	0.1802	7	0.291783
B_rodentium	Yes	0.005797	0.001289	0.01722	-0.01594	0.004072	3.914	7	0.005101
A_crotonat oxidans	No	0.351546	0.01404	0.02002	-0.005975	0.005988	0.9979	7	0.15468
R_gnavus	Yes	0.010574	0.007389	0.02422	-0.01683	0.004866	3.458	7	0.007754
B_sartorii	Yes	0.000112	0.004167	0.03875	-0.03458	0.004466	7.744	7	0.000247
P_buccalis	Yes	0.043903	0.004991	0.04197	-0.03698	0.01508	2.453	7	0.021463
B_coccoides	No	0.735221	0.05906	0.04862	0.01044	0.02968	0.352	7	0.269581
P_goldsteini	Yes	0.000048	0.3438	0.08065	0.2632	0.0298	8.83	7	0.000212
B_acidifaciens	Yes	0.0003	0.03951	0.1153	-0.07577	0.01145	6.616	7	0.00033
B_massiliensis	Yes	0.000236	0.02073	0.2286	-0.2079	0.03023	6.877	7	0.00033
A_muciniphila	Yes	0.032709	0.05799	0.2367	-0.1787	0.06732	2.655	7	0.01799

Supplementary table2 (New)

	Discovery?	P value	Mean of Insulin	Mean of Phlorizin	Difference	SE of difference	t ratio	df	q value
A_crotonat oxidans	No	0.232727	-0.7333	0.03382	-0.7672	0.5781	1.327	6	0.3072
O_sinus	No	0.575557	-0.7543	-1.189	0.4347	0.7345	0.5919	6	0.542668
B_sartorii	Yes	0.012428	-1.361	-0.2432	-1.118	0.317	3.526	6	0.041014
R_gnavus	No	0.062587	-1.523	-0.3416	-1.181	0.5176	2.282	6	0.137691
B_massiliensis	Yes	0.00827	-1.642	-0.3966	-1.246	0.3219	3.87	6	0.041014
B_coccoides	No	0.298653	-1.687	-1.447	-0.2409	0.2117	1.138	6	0.328518
J_ignava	No	0.954388	-2.898	-2.859	-0.0391	0.6557	0.05963	6	0.78737
B_hansanii	No	0.183217	-2.998	-1.793	-1.205	0.801	1.504	6	0.302308

Supplementary table 3 (New)

	Discovery?	P value (Mann-Whitney)	q value
Enterobacter Enterobacter_tabaci(NR_146667.1)	No	0.097781	0.654977
Megasphaera Megasphaera_elsdenii(NR_102980.1)	No	0.925503	>0.999999
Megamonas Megamonas_funiformis(AB300988)	No	0.304438	0.78437
Klebsiella Klebsiella_variicola(AJ783916)	No	0.00774	0.178793
Prevotella Prevotella_copri(AB064923)	No	0.496582	0.94184
Phascolarctobacterium Phascolarctobacterium_faecium(X72865)	No	0.098297	0.654977
Bacteroides Bacteroides_vulgatus(CP000139)	No	0.14177	0.654977
Bacteroides Bacteroides_caccae(X83951)	No	0.735423	0.943793
Bacteroides Bacteroides_thetaiotaomicron(AE015928)	No	0.197368	0.759017
Clostridium_XVI Clostridium_... (NR_147306.1)	No	0.611004	0.84184

Supplementary table 7 (New)

**For instance, the term "dysbiosis" is under extensive controversy. Wat would be dysbiosis here ?*

Truly, we need to be careful when we use the term, "dysbiosis". Here in this manuscript, major characteristics of "dysbiosis" was the induced proportion of *B. mass*. Indeed, primary component analysis suggested that it could be a promising candidate and specific analysis confirmed it as above. In the revised manuscript, we clearly stated this point as follows, "The relevance of the data from PCoA and individual analyses suggest that these bacteria could be good marker for dysbiosis in this model mice. ("Insulin suppressed the accumulation of potently oncogenic bile acids through changes in microbiome in STAM mice" in "Results")."

** The results of the fecal microbiota transplantation performed were essentially the same compared with Antibiotic treatment alone given that implantation of exogenous microbiota (precisely the Bacteroides anaerobic bacteria) was mainly unexplored. A more extensive germ-free model would help here in providing more evidences.*

Thank you very much for the useful comment. As reviewer #4 pointed out, whether implantation of specific bacteria (the suspected candidate in this study, *B. mass*) to germ-free STAM mice results in promotion of tumor-development would clarify the causal relationships between *B. mass* and development of liver tumor.

However, it is very difficult to isolate and cultivate *B. mass* because of its strictly anaerobic character. Therefore, we decided to conduct gut sterilization and FMT, respectively, both of which resulted in significant suppression of tumor-development in STAM mice (**Fig. 6**). By the results from FMT from normal mice, resident floral bacteria in normal intestinal environment could be excluded from candidates which were responsible for pathogenesis in this model. On the other hand, the results from FMT from STAM mice supported the validity of this experiment. Although causal relationships should be proved by implantation of *B. mass* to germ-free STAM mice, the only detectable bacteria which explain these results in an integrated way.

Fig 6

**Identical arguments could be given in the microbiota alterations found in the intestinal insulin receptor knockout mice.*

Thank you very much for the useful comment. To clearly conclude the causal relationship between *B. mass* and development of liver tumor in this model, isolation/implementation of *B. mass* to germ-free insulin-treated-ieIRKO-STAM mice is desirable, but as described above, it is extremely difficult.

In this study, in the experiment with ieIRKO-STAM mice, we succeeded to prove the loss of

intestinal epithelial insulin receptor signaling resulted in suppression of the expression of AMPs (antimicrobial peptides) and improvement of dysbiosis (characterized by the proportion of *B. mass*, as was discussed in this rebuttal letter), which was achieved by insulin supplementation in STAM mice.

We hope that we will be able to conduct with isolation and implementation techniques for *B. mass* and prove the causal relationship between *B. mass* and development of liver-tumor in the near future.

**Insulin Glargine has been described to change endogenous insulin receptors and IGF-I in the liver or improve several NAFLD phenotypes in humans. This was not mentioned in the manuscript.*

Thank you very much for the comment. As reviewer #4 commented, insulin glargine has been reported to suppress liver phenotypes in diabetic NAFLD patients in several human studies (Diabetes Metab Res Rev. 2020;36:e3292, Obes Metab. 2016;18(Suppl 2):50-58., Diabetes Ther. 2018;9(3):1253-1267., Diabetes Care. 2015;38(7):1339-1346). However, it is expected to be a common effect of insulin *per se* because intensive insulin therapy (Diabetes Metab Res Rev. 2014;30(6):521-529.), compounding insulin (Mix25; Acta Diabetol. 2014 Oct;51(5):865-73.), combinational therapy (Met 1000mg + NovoLog Mix 70/30; J Investig Med. 2012 Oct;60(7):1059-63. BiAsp 30 insulin and metformin; J Diabetes Complications. 2007; 21(3): 137–142.) also improved NAFLD phenotypes in humans as was explained in previous review article (Diabetologia volume 64, pages1461–1479 (2021)).

In the section titled “Insulin treatment suppressed NASH in STAM mice at multiple steps” in “Results”, we added words to the manuscript and inserted the references as follows.

“These data suggest that insulin ameliorates hepatic steatosis by preserving fat storage in white adipocytes under diabetic conditions, consistent with previous clinical studies (Diabetologia volume 64, pages1461–1479 (2021))”.

As reviewer #4 pointed out, previous study reported that insulin glargine promoted the expression of *Igf-1r*, *Insr* genes in liver of diabetic rats, which could overstimulate the mitogenic signaling pathways while the expression of *Igf-1* decreased (Iran J Basic Med Sci. 2018 May; 21(5): 489–494.). In our experiment, we supplemented systemic insulin signal to keep them in normoglycemia. Therefore, the expression level of *Igf-1* and *Igf-1r* could change hepatic mitogenic signal in relation to IR/IGF1-R signaling cascade. However, the expression level of *Igf-1* and *Igf-1r* was not changed significantly between the groups by drug administration (glargine or phlorizin. The data are shown below in addition to **Fig. S3J**).

In “Data Accessibility”, we added accession number as follows, PRJNA866345 (SRA, <https://dataview.ncbi.nlm.nih.gov/object/PRJNA866345?reviewer=egt5015oc8n6ir16q44ton2m>dv), GSE210876 (GEO, <https://www.ncbi.nlm.nih.gov/geo/query/acc.cgi?acc=GSE210876>), GSE210517 (GEO, <https://www.ncbi.nlm.nih.gov/geo/query/acc.cgi?acc=GSE210517>). The access token is required to download the latter two data sets for peer review. For GSE210876 (GEO) and GSE210517 (GEO), `sxexuismpxsldof` and `wzktgcmxfybdgd` should be fulfilled respectively. They will be available publicly until publication. For human 16S metagenomics, the data is available at DDBJ web page, Study: JGAS000574, Dataset: JGAD000700.

REVIEWER COMMENTS

Reviewer #2 (Remarks to the Author):

In my last review feedback, I pointed out that there are several important issues (6 questions) from the previous comments (first round review) remained unaddressed. I stated that additional experiments are necessary and the results to be generated are important to support the proposed mechanisms and the conclusions. Unfortunately, in the second revision, these questions are still open. No additional work has been done to address any of these issues. Therefore, my views remain the same – I'm not convinced that this work makes significant advances to the field and adds to the current understanding of NASH/HCC development.

Reviewer #4 (Remarks to the Author):

I appreciate the significant effort dedicated to answer the comments.

An ANCOM or Deseq analysis to explore the human microbiota compositional nature is still missing. PCoA seems not enough in these cases.

The elevated number of "ND" results in Supplementary table 5 (especially for cholic and deoxycholic acid), even including one subject with virtually no information is worrying.

Trying to justify sample size in humans with animal results is, at least, atypical.

It is also not usual to merge the abundance of three different species to obtain a q value in Supplementary Table 3.

Reviewer #2

In my last review feedback, I pointed out that there are several important issues (6 questions) from the previous comments (first round review) remained unaddressed. I stated that additional experiments are necessary and the results to be generated are important to support the proposed mechanisms and the conclusions. Unfortunately, in the second revision, these questions are still open. No additional work has been done to address any of these issues. Therefore, my views remain the same – I'm not convinced that this work makes significant advances to the field and adds to the current understanding of NASH/HCC development.

Thank you very much for the comments. The comments from reviewer #2 were very much useful to improve the manuscript by making the limitations clearer than the initial one. The editor also added the comments as follows: “In addition, please provide the functional analysis of metagenomic data requested by R#2 in point 6, or if this is technically infeasible to do, please explain why and include a caveat in the text. (*)”

We tried to conduct additional experiments as reviewer #2 recommended. However, there are technical difficulties in demonstrating function of specific bacteria in this study. Especially it is difficult to generate STAM mice in germ-free conditions. First, the vulnerability in the general feature of germ-free mice would make it difficult to administrate toxic amounts of STZ for pancreatic β cells in their neonate phase. Second, hygienic control is hardly achieved in vinyl isolators by extreme polyurea and high-fat-diet feeding to generate STAM mice, because cages for these mice are usually needed to clean up three times a week.

Third, as we already addressed in the original manuscript, a method to isolate and culture *B. mass* has not been established, and thus it was impossible for us or our collaborators to amplify from the stool of STAM mice sufficiently to inoculate other recipient mice or to exert functional analysis. Indeed, we found no hits when we searched in PubMed (https://pubmed.ncbi.nlm.nih.gov/?term=bacteroides+massiliensis+culture&filter=pubt.review&filter=datesearch.y_5), and *B. mass* is not commercially available to our knowledge.

Thus, we decided to describe the caveats of this study regarding these points sufficiently, following the suggestions by the editor.

Moreover, this study revealed the protective effects of intestinal epithelial insulin signal against developing liver tumor in DM-NASH model mice. Although it was supported mainly by *in vivo* experiments, this is the first report that insulin supplementation inhibits tumor development via intestinal epithelium in model mice. The concept in this study offers the basic evidence for non-obese diabetic NASH patients that insulin analogues are also a safe option.

Taken together, we believe the novelty would be found in this study, shedding light on the

protective role of intestinal insulin signal against tumor development in NASH comorbid with diabetes, even though further studies are needed to demonstrate the mechanism related to microorganisms in the future.

In the revised manuscript, we further clearly stated the limitations of this study as follows.

[Introduction, the last paragraph]

Here, we demonstrate the protective role of insulin signaling in the gut against hepatocarcinogenesis associated with maintaining intestinal barrier function and suppressing dysbiosis using diabetic NASH-HCC model mice and samples from patients with diabetes and NASH.

[Results, § Insulin suppressed the accumulation of potentially oncogenic bile acids associated with changes in the microbiome in STAM mice, the second paragraph]

The bacterium was reported to have bile acid hydrolase (BSH) responsible for the deconjugation of taurine- or glycine-conjugated primary bile acids, based on the database (PATRIC 3.6.12, *Bacteroides massiliensis* B84634 = Timone 84634 = DSM 17679 = JCM 13223), and thus could be involved in supplying a substrate for secondary bile acids production, although it has not been experimentally confirmed because of the difficulty in inoculation of the bacteria.

[Results, § Patients with diabetes and NASH showed a similar signature of gut flora to that of STAM mice which was preserved by insulin treatment, the last paragraph]

Although a significant change was not observed in Analysis of Compositions of Microbiomes with Bias Correction 2 (ANCOM-BC2³⁰, Fig. S8A, Supplementary Table 7) and *Bacteroides massiliensis* itself (Fig. S8B, S8C), the abundance of another genetically similar bacterium, *Bacteroides vulgatus*, changed significantly (Fig. 5D, unpaired t test). The bacterium was focused on because we could refer to the previous reports, which suggest the relevance between the abundance of genus *Bacteroides* and insulin-use³¹, or show genetic similarity between *Bacteroides vulgatus* and *Bacteroides massiliensis*³². In addition, the abundance of *Bacteroides vulgatus* was also known to increase in the gut flora of fatty liver patients^{33 34}. Host preference is also assumed between mice and humans.

[Results, § Antibiotic treatment or fecal microbiota transplantation suppressed the development of HCC in STAM mice, the second paragraph]

In the antibiotics-treated group, the number of visible tumors (Fig. 6C, 6D) and their maximum diameter (Fig. 6E) were significantly suppressed, supporting the contribution of

the certain bacteria in the gut flora of STAM mice.

[Results, § Antibiotic treatment or fecal microbiota transplantation suppressed the development of HCC in STAM mice, the third paragraph]

Next, we carried out fecal microbiota transplantation (FMT), considering the difficulty of inoculation because of the strict anaerobe of *B. massiliensis*.

[Results, § Antibiotic treatment or fecal microbiota transplantation suppressed the development of HCC in STAM mice, the fourth paragraph]

These data suggest that certain bacteria, usually suppressed in normal flora, promoted the development of tumor in the liver in STAM mice.

[Results, § Antibiotic treatment or fecal microbiota transplantation suppressed the development of HCC in STAM mice, the sixth paragraph]

These data support the possibility that certain bacteria including *B. massiliensis* promoted liver tumor development in this model, although gnotobiotic experiments are needed to demonstrate specific mechanisms.

[Discussion, the fifth paragraph]

In the current study, we have also found that HFD feeding and STZ injection led to dysbiosis associated with changes in secondary bile acids, and thus dysbiosis might play a causal role in the development of HCC, although it remains undetermined exactly which bacteria play a role in this model.

[Discussion, the sixth paragraph]

In this study, it remains undetermined that *B. massiliensis* plays a role in promoting HCC in this model. Although isolation and implantation to germ-free mice are needed to assert robust conclusions, it is extremely difficult to isolate this specific bacterium. The vulnerability of germ-free mice and hygienic control against polyurea and high fat diet feeding to generate STAM mice in germ-free isolators are other technical issues to conquer. Many clinical parameters also remain unknown in the human study. Thus, further efforts using animal models are needed, and the collection of various specimens and clinical data from patients with NASH and diabetes are also required to clarify the direct relationship between the specific species and HCC development in the future.

Reviewer#4

I appreciate the significant effort dedicated to answer the comments.

Thank you for the positive comment on our revised version of manuscript.

An ANCOM or Deseq analysis to explore the human microbiota compositional nature is still missing. PCoA seems not enough in these cases.

Thank you very much for the useful comment. In this study, we found that insulin administration suppressed the abundance of *Bacteroides massiliensis* (*B.mass*) in the stool of STAM mice, which suggested the possibility of a similar phenomenon in human patients even though there might be a difference in colonization preference between the hosts. Referring to the previous report that the relative abundance of the genus *Bacteroides* was suppressed in insulin-users¹ in patients with diabetes even though they were type 1 DM, we planned to focus on the changes in *Bacteroides* species.

As a result, we found that the relative abundance of *B.mass* was not suppressed by insulin treatment, presumably because the average occupancy of *B.mass* was only up to 0.01% in NASH patients without insulin-use, which was much lower than STAM mice (**Fig.S8B, S8C**). On the other hand, the genetically similar² bacterium, *Bacteroides vulgatus*, was significantly suppressed in patients receiving insulin treatment (by unpaired t-test, shown in the revised version of **Figure 5D**). These results suggest the relevancy between insulin treatment and certain species in genus *Bacteroides* in mice and humans.

As a non-biased strategy, we also conducted differential analysis by ANCOM, as reviewer #4 recommended. Although we could not find bacteria that showed a statistical significance, a relatively large fold change with a difference close to the statistical significance was found in *B. vulgatus* even in broader spectrum of bacteria (p value, **Fig.S8A**), which was not limited to the genus *Bacteroides*. It is warranted that future study is conducted in a larger population. Figures and tables were revised along with the refinement of differential analysis by ANCOM (**Fig. S8A, Supplementary Table 7**).

Fig. 5D

Fig. S8A

taxon	Fold change (Natural Logarithm)	[-log10 p value]	[-log10 q value]
Bacteroides Bacteroides_uniformis(AB050110)	-1.28065694	0.480456195	0
Bacteroides Bacteroides_fragilis(CR626927)	0.010968259	0.002203529	0
Enterobacter Enterobacter_tabaci(NR_146667.1)	-3.124653512	1.042578193	0
Klebsiella Klebsiella_variicola(AJ783916)	-3.939992378	2.090698161	0
Bacteroides Bacteroides_vulgatus(CP000139)	-2.246559113	0.985761217	0
Bacteroides Bacteroides_dorei(AB242142)	-0.720403034	0.203555751	0
Alistipes Alistipes_onderdonkii(AY974071)	-1.402169225	0.443895356	0
Parabacteroides Parabacteroides_distasonis(AB238922)	-0.524395856	0.148877188	0
Akkermansia Akkermansia_muciniphila(AY271254)	1.323291277	0.36260933	0
Faecalibacterium Faecalibacterium_prausnitzii(AJ413954)	0.778709913	0.907268913	0
Bacteroides Bacteroides_thetaiotaomicron(AE015928)	-1.519860218	0.646641553	0
Bacteroides Bacteroides_ovatus(AB050108)	-0.679220854	0.165796408	0
Bacteroides Bacteroides_caccae(X83951)	-0.603644086	0.202863328	0
Parabacteroides Parabacteroides_merdae(AB238928)	-1.515700133	0.501682925	0
Clostridium_XIVa Lachnospirillum_pacaense(NR_147396.1)	-0.435752328	0.110220902	0
Alistipes Alistipes_senegalensis(NR_118219.1)	-1.10943633	0.345893397	0
Collinsella Collinsella_aerofaciens(NR_113316.1)	-1.490094846	0.452828241	0
Prevotella Prevotella_copri(AB064923)	-1.210336309	0.376881775	0
Phascolarctobacterium Phascolarctobacterium_faecium(X72865)	-2.181269123	0.835991318	0
Megasphaera Megasphaera_elsdenii(NR_102980.1)	-1.088325808	0.341088135	0

Supplementary Table 7

The elevated number of "ND" results in Supplementary table 5 (especially for cholic and deoxycholic acid), even including one subject with virtually no information is worrying.

We apologize for the insufficiency of metadata in human observational study in the previous version of the manuscript, and we now fulfilled Pt#25 reinvestigated the original record's clinical data (**Supplementary Table 4**). There are no significant differences in any clinical parameters between the insulin-treated and non-insulin-treated groups (**Supplementary Table 6**).

The population of this study is prepared for multiple use to explore the feature of NASH patients by various methods. In an animal experiment, we have found it important to preserve intestinal insulin signal to protect from the pathogenesis of NASH-HCC. These results prompted us to confirm the hypothesis with available clinical samples. In the future, we will prepare a new cohort, securing clinical background data about glucose tolerance and bile acids.

24	nonDM		4	2	73	31.6	112	106	200	72
25	DM/Ins(-)	NASH_LC	1	90	17.9	69	56	128	130	
26	DM/Ins(-)		4	1	72	26.6	35	22	192	102
27	DM/Ins(+)	NASH_LC	2	69	23.0	27	13	161	68	

(Supplemental Table 4)

	Insulin treatment	Age	BMI [cm/m ²]	AST [IU/L]	ALT [IU/L]	TC [mg/dL]	TG [mg/dL]
Average	No insulin use	73.13333	28.67333	42.06667	38.06667	175.7857	118.7143
	Insulin use	65.6	31.56	61.8	61.6	157.2	95.6
Standard Error	No insulin use	2.922235	1.507277	7.648173	7.159599	10.09785	12.63012
	Insulin use	4.132191	8.164068	36.7141	46.05242	6.318623	21.27675
N	No insulin use	15	15	15	15	14	14
	Insulin use	5	5	5	5	5	5
p value		0.17772	0.549502	0.353287	0.360329	0.191748	0.373174

(Supplemental Table 6)

Trying to justify sample size in humans with animal results is, at least, atypical.

Although there were few studies about metagenomic change by insulin therapy, we could refer to the observational study in which the metagenomic difference between insulin-user and non-insulin-user¹. They compared insulin-user (n = 13) with non-insulin-user (n = 8) to find a statistically significant difference in relative abundance of the genus *Bacteroides*. In addition, some studies related to GLP1 receptor agonists were also conducted with the sample size of dozens^{3,4} to find out gut floral differences.

Moreover, as we presented in point-by-point discussion previously, the size (n = 20) in this pilot study was also reasonably supported by the results of our *in vivo* experiment. These data led us to conduct 16S metagenomics when 20 samples were prepared for analysis.

It is also not usual to merge the abundance of three different species to obtain a q value in Supplementary Table 3.

We appreciate the advice from reviewer #4. We found statistical significance between the two groups from the analysis of *Bacteroides vulgatus* (Fig.5D). Considering the skewness of the distribution, it was reasonable to assess the logarithm of the relative abundance of *B. vulgatus* in the stool of human patients.

Fig.5D (Notified again in this point-by-point discussion)

Reference

1. Sci Rep. 2014 Jan 22;4:3814. doi: 10.1038/srep03814. Fecal microbiota imbalance in Mexican children with type 1 diabetes
2. Int J Syst Evol Microbiol 55, 1335-1337, doi:10.1099/ijs.0.63350-0 (2005). doi: 10.1099/ijs.0.63350-0. *Bacteroides massiliensis* sp. nov., isolated from blood culture of a newborn
3. Front Endocrinol (Lausanne). 2022 Jan 10;12:814770. doi: 10.3389/fendo.2021.814770. Gut Microbial Signatures for Glycemic Responses of GLP-1 Receptor Agonists in Type 2 Diabetic Patients: A Pilot Study
4. Endocrinol Diabetes Metab. 2017 Dec 28;1(1):e00009. doi: 10.1002/edm2.9. Gut microbiome differences between metformin- and liraglutide-treated T2DM subjects

REVIEWERS' COMMENTS

Reviewer #4 (Remarks to the Author):

I have no further comments.

Reviewer #4 (Remarks to the Author):

I have no further comments.

We are grateful to your favorable comment. Thank you.